# Impacts of Enhanced Weathering on biomass production for negative emission technologies and soil hydrology

Wagner de Oliveira Garcia*[1], Thorben Amann[1], Jens Hartmann[1], Kristine Karstens[2], Alexander Popp[2], Lena R. Boysen[3], Pete Smith[4], Daniel Goll[5,6]

[1]Institute for Geology, Center for Earth System Research and Sustainability, Universität Hamburg, Germany
[2]Potsdam Institute for Climate Impact Research (PIK), Germany
[3]Max Planck Institute for Meteorology, Germany
[4]Institute of Biological and Environmental Sciences, School of Biological Sciences, University of Aberdeen
[5]Laboratoire des Sciences du Climat et de l'Environnement, CEA CNRS UVSQ, 91190 Gif-sur-Yvette, France
[6]Institute of Geography, University Augsburg, Germany

*Correspondense to: Wagner de Oliveira Garcia (wagner.o.garcia@gmail.com) ORCID: https://orcid.org/0000-0001-9559-0629

**Abstract.** Limiting global mean temperature changes to well below 2°C likely requires a rapid and large-scale deployment of Negative Emission Technologies (NETs). Assessments so far showed a high potential for biomass based terrestrial NETs, but only few included effects of the commonly found nutrient deficient soils on biomass production. Here, we investigate the deployment of Enhanced Weathering (EW) to supply nutrients to phosphorus (P) deficient areas of Afforestation/Reforestation and naturally growing forests (AR) and bio-energy grasses (BG), besides the impacts on soil hydrology. Using stoichiometric ratios and biomass estimates from two established vegetation models, we calculated the nutrient demand of AR and BG. Insufficient geogenic P supply limits C storage in biomass. For a mean P demand by AR and a low geogenic P supply scenario, AR would sequester 119 Gt C in biomass; for high geogenic P supply and low AR P demand scenario, 187 Gt C would be sequestered in biomass; and for a low geogenic P supply and high AR P demand, only 92 Gt C would be accumulated by biomass. An average amount of ~150 Gt basalt powder applied for EW would be needed to close global P gaps and completely sequester projected amounts of 190 Gt C during years 2006 – 2099 for the mean AR P demand scenario (2 – 362 Gt basalt powder for the low AR P demand and for the high AR P demand scenarios would be necessary respectively). The average potential of carbon sequestration by EW until 2099 is ~12 Gt C (~0.2 – ~27 Gt C) for the specified scenarios (excluding additional carbon sequestration via alkalinity production). For BG, 8 kg basalt $m^{-2}$ $a^{-1}$ might, on average, replenish the exported potassium (K) and P by harvest. Using pedotransfer functions, we show that the impacts of basalt powder application on soil hydraulic conductivity and plant available water, to close predicted P gaps, would depend on basalt and soil texture, but in general the impacts are marginal. We show that EW could potentially close the projected P gaps of an AR scenario, and exported nutrients by BG harvest, which would decrease or replace the use of industrial fertilizers. Besides that, EW ameliorates the soils capacity to retain nutrients and soil pH, and replenish soil nutrient pools. Lastly, EW application could

improve plant available water capacity depending on deployed amounts of rock powder - adding a new dimension to the coupling of land-based biomass NETs with EW.

## 1. Introduction

To limit temperature increase due to climate change to well below 2°C compared to pre-industrial levels by the end of the century, research efforts on negative emission technologies (NET; i.e., ways to actively remove $CO_2$ from the atmosphere), intensify. Terrestrial NETs encompass, Bioenergy with Carbon Capture and Storage (BECCS), Afforestation, Reforestation and natural growing forests (AR), Enhanced Weathering (EW), Biochar, restoration of wetlands, and Soil Carbon Sequestration. From these land-based NET options, BECCS, AR, Biochar, and EW can potentially be combined for increasing atmospheric carbon dioxide removal (CDR) (Smith et al., 2016;Beerling et al., 2018;Amann and Hartmann, 2019).

BECCS combines energy production from biomass and carbon capture at the power plant with subsequent storage. Sources for biomass-based energy production are crop and forestry residues (Smith, 2012;Smith et al., 2012;Tokimatsu et al., 2017), dedicated bio-energy grass (BG) plantations (Smith, 2012;Smith et al., 2012) or short rotation woody biomass from forestry (Cornelissen et al., 2012;Smeets and Faaij, 2007). Large-scale AR, as well as bio-energy plantations, require extensive landscape modifications for growing forests or natural regrowth of trees in deforested areas to increase terrestrial CDR (Kracher, 2017;Boysen et al., 2017a;Popp et al., 2017;Humpenöder et al., 2014), and huge quantities of irrigation water (Boysen et al., 2017b;Bonsch et al., 2016). The biomass yields of AR and agricultural bio-energy crops directly correlate with fertilizer application, which in turn could reduce CDR efficiency due to related emissions of $N_2O$ (Creutzig, 2016;Popp et al., 2011) and initiate unwanted side-effects like acidification of soils (Rockström et al., 2009;Vitousek et al., 1997), streams/rivers, and lakes (Vitousek et al., 1997).

Under intensive growth scenarios, nutrient supply is a critical factor. According to Liebig's law of the minimum, supplying high amounts of nitrogen (N) might shift growth limitation to other nutrients (von Liebig and Playfair, 1843). Some U.S. forests already show changes from N-limited to a Phosphorus (P) limited system caused by increases in N atmospheric deposition (Crowley et al., 2012) along with magnesium (Mg), potassium (K) and calcium (Ca) deficiencies (Garcia et al., 2018;Jonard et al., 2012). Poor nutrient supply, related to deficient mineral nutrition, may reduce tree growth (Augusto et al., 2017). Impacts on biomass production due to poor tree nutrition is observed in European forests (Knust et al., 2016;Jonard et al., 2015) decreasing the carbon sequestration of forest ecosystems (Oren et al., 2001) – a factor rarely included in climate models leading to overestimated CDR potentials.

Specifically, global simulations with a N-enabled land surface model (Kracher, 2017) suggest that insufficient soil nitrogen availability for a RCP4.5 AR scenario (Thomson et al., 2011) could lead to a reduction in the cumulative forest carbon sequestration between year 2006 – 2099 by 15%. Goll et al. (2012) showed that carbon sequestration during the 21st century in the JSBACH land surface model was 25% lower when N and P effects were considered.

Mineral weathering is a natural and primary source of geogenic nutrients (i.e., Mg, Ca, K, P, etc.; Hopkins and Hüner, 2008;Landeweert et al., 2001;Waldbauer and Chamberlain, 2005;Singh and Schulze, 2015), and controls atmospheric $CO_2$ concentrations over geological timescales (Walker et al., 1981;Lenton and Britton, 2006;Berner and Garrels, 1983;Waldbauer and Chamberlain, 2005;Yasunari, 2020). Chemical dissolution of silicate minerals increases alkalinity fluxes (Kempe, 1979;Gaillardet et al., 1999;Hartmann et al., 2009), and natural weathering sequestration can range from 0.1 to 0.3 Gt C $a^{-1}$ (Gaillardet et al., 1999;Moon et al., 2014;Hartmann et al., 2009). To sequester significant amounts of $CO_2$ within decades, EW aims to speed up weathering processes by increasing the mineral reactive surfaces through rock comminution (Hartmann et al., 2013;Schuiling and Krijgsman, 2006;Hartmann and Kempe, 2008). Mineral-soil-microorganism interactions (e.g., by mycorrhizal fungi; Kantola et al., 2017;Landeweert et al., 2001;Taylor et al., 2009) increase the volume of soil that plant roots can extract nutrients from (Clarkson and Hanson, 1980;Hopkins and Hüner, 2008), which might enhance the weathering activity in addition to the reaction with dissolved $CO_2$. EW further increase soil pH by alkalinity fluxes, and could be a long-term source of macro- (e.g., Mg, Ca, K, P, and S), and micronutrients (e.g., B, Mo, Cu, Fe, Mn, Zn, Ni) (Leonardos et al., 1987;Nkouathio et al., 2008;Beerling et al., 2018;Hartmann et al., 2013;Anda et al., 2015) rejuvenating the nutrient pools of soils.

P is rather immobile soil nutrient and only a small fraction of soil P is readily available for plant uptake limiting plant growth in a wide range of ecosystem (Shen et al., 2011;Elser et al., 2007). P content in soils is a result of a process controlled by the interactions of parent material (primary rocks) with climate, tectonic uplift, and erosion history through geological time (Porder and Hilley, 2011). The processes of P transfer between biologically available and recalcitrant P pools influence at most P availability (Porder and Hilley, 2011). Orthophosphate ($H_2PO_4^-$ or $HPO_4^{2-}$) is the chemical species adsorbed by plants (Shen et al., 2011) and its solubility is controlled by soil pH as de-protonation occurs when pH increases. Ideal pH conditions for orthophosphate availability are from 5 to 8 (Holtan et al., 1988) with soil moisture influencing soil P availability for different crops (He et al., 2005;He et al., 2002;Shen et al., 2011), and natural ecosystems (Goll et al., 2018).

The inclusion of soil hydraulic properties in the evaluation of EW effects is important as the soil water content has a strong influence on average crop yield. Practices that increase the plant available water (PAW) are thought to mitigate drought effects on crops (Rossato et al., 2017). The water content of soils also seems to influence soil erosion rates and surface runoff (Bissonnais and Singer, 1992). In addition, soil water content influences soil $pCO_2$ production, which is a relevant agent for mineral dissolution (Romero-Mujalli et al., 2018).

Deploying land-based NETs would imply large changes in a local landscape nutrient and water cycles. At least 65% of worldwide soils (6.8 billion hectares of land) have unfavorable soil conditions for agricultural production (Fischer et al., 2001). Therefore, we assess if applications of rock mineral based P sources could close eventual nutritional gaps in an environment with natural N supply (N-limited) and with N fertilization (N-unlimited), using a global afforestation scenario. In addition, we investigate the effects of coupling nutrient supplying (EW) to nutrient demanding (AR and BG) land-based NETs by focusing on the efficiency of different upper limits of basalt powder to supply nutrients. We hypothesize that large-scale EW deployment

potentially changes soil texture. Therefore, threshold values for impacts on soil hydraulic conductivity, and plant available water will be determined.

## 2. Methods

Since phosphorus (P) is a limiting nutrient in a wide range of ecosystems (Elser et al., 2007), we performed a P budget for an N stock-based P demand from an AR scenario considering natural N supply (hereafter N-limited) and N fertilization (hereafter N-unlimited). We selected two N supply scenarios since the related P demand is proportional to biomass N stock, but in the main text we discuss only the N-limited AR scenario. We estimated the balanced supply of Mg, Ca, and K for each supplied P based on ideal Mg, Ca, and K demand of AR derived from databases of biomass nutrient content. Balanced nutrient supply is necessary to avoid shift of growth limitation to other nutrients, which can occur according to Liebig's Law (von Liebig and Playfair, 1843). Shift of growth limitation to other nutrients is observed for some U.S. forests that changed from a N-limited to a Phosphorus (P) limited after increase in atmospheric N deposition (Crowley et al., 2012). Based on minimum and maximum harvest rates of bio-energy grass (BG), we estimated the related P and K export by harvest from the fields. We decide on these nutrients for BG since crops require large amounts of K and P, once N demand is covered. The amount of rock powder for Enhanced Weathering (EW) to cover projected P gaps and to replenish exported nutrients was estimated. The projected impacts on soil hydrology due to EW deployment were done by pedotransfer functions since they are used to estimate soil hydraulic properties (Schaap et al., 2001;Whitfield and Reid, 2013;Wösten et al., 2001) and such approximations have proven to be a suitable approach (Vienken and Dietrich, 2011).

The additional AR P demand, obtained for the 21st century for an N-unlimited and N-limited AR scenario (Kracher, 2017) was approximated by stoichiometric P:N ratios for mean and range (5th and 95th percentiles), which is a similar approach done by Sun et al. (2017). The ratios were derived from databases of hard- and softwood (Pardo et al., 2005) and foliar biome-specific nutrient content (Vergutz et al., 2012). We then compared the inferred P demand to geogenic P supply given by observation-based estimates of soil inorganic labile P and organic P (Yang et al., 2014a), observation-based estimates of P release (Hartmann et al., 2014) from weathering corrected to future temperature increase, since the uncertainty on future hydrological cycle is too high (Goll et al., 2014) and estimated atmospheric P depositions from Wang et al. (2017) to derive the potential geogenic P deficits (i.e, the P gap) during the 21st century. Since the geogenic P supply cannot cope with N stock-based P demand from the different AR scenarios within P gapped areas, the biomass production and biomass C sequestration, predicted by the AR scenarios, will be lower. Based on the amount of missing P, we estimated the C-stock reduction within P gapped areas by using stoichiometric C:P ratios. The C:P ratios were derived from simulated C stock content (Kracher, 2017) and inferred N stock-based P demand.

Necessary Mg, Ca, and K supply for balanced tree nutrition based on P supply were derived from N stock-based Mg, Ca, and K additional demand normalized to the N stock-based additional P demand (Fig. 1). The nutrient demand of bio-energy grass

was estimated based on stoichiometric P:N and K:N ratios, used in Bodirsky et al. (2012), for minimum and maximum exported N proportional to harvest rates of the 1995 – 2090 period obtained from the agricultural production model MAgPIE (Fig. 1). Later on, the necessary amount of rock to cover the P gaps of AR scenario and to resupply the exported nutrients by BG harvest was estimated (Fig. 1). In addition, the potential impact of deploying rock powder into the topsoil hydrology was done.

## 2.1. Global land-system model output

### 2.1.1. Afforestation/Reforestation

The idealized simulations for the AR system from Kracher (2017) performed by the land surface model JSBACH (Reick et al., 2013) for a representative greenhouse concentration pathway 4.5 (RCP4.5) were used (Thomson et al., 2011). The RCP4.5 scenario assumes that the emissions peak is around 2040 and considers that forest lands expand from their present day extent (Thomson et al., 2011). The coupled terrestrial nitrogen-carbon cycle model assumes N-unlimited and N-limited conditions, and consider harvest rates, and transitions between different anthropogenic and natural land cover types (Hurtt et al., 2011) for a Gaussian grid of approximately 2°×2° resolution. Accounting for the N cycle reduces the uncertainty on atmospheric carbon sequestration prediction by AR models (Zaehle and Dalmonech, 2011). In JSBACH, the N supply for plants is controlled by competition between plants and decomposing microbes, while other numerical models prioritize immobilization or plant growth (Achat et al., 2016).

The net primary productivity (NPP) calculation was based on atmospheric $CO_2$ concentrations, stomatal conductance, and water availability. JSBACH considers mass conservation, a supply-demand ansatz, and fixed C:N ratios (Goll et al., 2012). The coupled terrestrial nitrogen-carbon cycle model was selected since it: (i) considered forest regrowth on abandoned croplands (which in the long term become acidic and consequently favor leaching of nutrients and heavy metals (Hesterberg, 1993)); (ii) considered natural shift in natural vegetation; (iii) considered a natural N supply scenario (N-limited) and a N fertilized scenario (N-unlimited); (iv) considered future $CO_2$ increase leading to $CO_2$ fertilization; and (v) explicitly consider large-scale afforestation.

We retrieved the annual changes in N and C content of different pools, i.e., Wood (above and below ground, also including litter) and foliar (above and below ground, also including litter) for temperate, cold, tropical, and subtropical climate growing forests and shrubs plant functional types for years 2006 – 2099 and annual model output.

### 2.1.2. Biomass production from bio-energy grass

Simulations of BG nutritional needs from the agricultural production model MAgPIE, a framework for modeling global land-systems (Dietrich et al., 2018;Lotze-Campen et al., 2008;Popp et al., 2010) were used. The objective of MAgPIE is to minimize total costs of production for a given amount of regional food, bio-energy demand and climate target (here RCP4.5, to keep correspondence to the AR simulations). In its biophysical core, the yields in the model are based on LPJmL (Bondeau et al.,

2007;Beringer et al., 2011;Müller and Robertson, 2013), a dynamic global vegetation model, which is designed to simulate vegetation composition and distribution for both natural and agricultural ecosystems.

At the starting point of the simulation, the LPJmL bio-energy grass yields have been scaled using agricultural land-use intensity levels (Dietrich et al., 2012) for different world regions accounting for the yield gap between potential and observed yields for the period 1995 – 2005. For the future yields (2005 – 2090), the development is then driven by investments into yield-increasing technologies (Dietrich et al., 2014) based on the socio-economic boundary conditions of the system.

The MAgPIE output had a frequency of 10 years and the global minimum, and maximum of each output year was taken to obtain the potential bio-energy grass minimum (0.7 kg m$^{-2}$ a$^{-1}$) and maximum (3.6 kg m$^{-2}$ a$^{-1}$) harvest rate for the simulation period for the areas with bio-energy plantations.

## 2.2. Nutrient demand

### 2.2.1. Afforestation/Reforestation

The P, Mg, Ca, and K additional demand is defined as the amount of P, Mg, Ca, and K needed to realize the state of ecosystem N variables in each grid cell and year according to JSBACH output (Fig. 1). It was estimated from the spatially explicit information on average forest N content of each stock and plant functional type for an N-unlimited, and an N-limited AR scenario from Kracher (2017). Since P limits forest growth in a wide range of ecosystems (Elser et al., 2007), we performed a P budget for each AR scenario. The ideal P, Mg, Ca, and K biomass additional demand were based on the difference in the simulated change in N pools at that time with respect to the simulation year of 2006 multiplied by their corresponding Mg:N, Ca:N, K:N, or P:N ratios ($r_{ij}$) and were calculated following Eq. (1):

$$\Delta M_{pool,i} = \sum_{j=1}^{n} \Delta N_{ij} \times r_{ij}, \tag{1}$$

where $\Delta M_{pool,i}$ [kg m$^{-2}$ a$^{-1}$] is the average N stock-based Mg, Ca, K, or P demand for a given time in the future simulation time range (2007 – 2099) within a cell for biome i. $\Delta N_{ij}$ [kg m$^{-2}$ a$^{-1}$] is the average N stock change of pool j. n is the number of N pools. The N pools considered are: Wood (above and below ground, including litter) and foliar (above and below ground, including litter).

The P, Mg, Ca, K, and N content of leaves obtained from a global leaf chemistry database (Vergutz et al., 2012) was used to derive the Mg:N, Ca:N, K:N, or P:N ratios (Table 1), which was already biome classified. For wood, the tree chemical composition database of US forests (Pardo et al., 2005) was used in order to derive the global ratios, which were assumed to represent the chemical composition of all biomes (Table 1).

The AR C content (Fig. 2) from Kracher (2017) and the resulting N stock-based Mg, Ca, and K demand were normalized by the N stock-based P demand to estimate the mean and range C:P, Mg:P, Ca:P, and K:P ratios of each grid cell. The

stoichiometric C:P, Mg:P, Ca:P, and K:P ratios were used to derive the C-fixation reduction due to P deficiencies and the necessary Mg, Ca, and K supply for a balanced biomass nutrition based on supplied P (Fig. 1).

### 2.2.2. Biomass production from bio-energy grass

The BG yield was obtained by the spatially explicit harvest rates within a grid cell for an output frequency of 10 years and a period of 95 years (1995 – 2090). The minimum 0.7 kg m$^{-2}$ a$^{-1}$ and maximum 3.6 kg m$^{-2}$ a$^{-1}$ harvest rate were used. With the information on exported N by each harvest rate, the exported K or P from cultivation fields (Eq. 2) were estimated based on the P:N, and K:N stoichiometric ratios used in Bodirsky et al. (2012). We have chosen these nutrients, since crops require large amounts of K and P.

The simulated forests from the AR scenario are perennial, differently from bio-energy grasses that are completely harvested regularly due to their use as biomass feedstock for BECCS. Thus, the natural system's nutrient supply is insufficient to maintain successive and constant yields, and the exported nutrients by harvest need to be replenished (Cadoux et al., 2012) to maintain high yields. The exported nutrients were calculated following Eq. (2):

$$Bio_x = r_x \times N_{harvest}, \tag{2}$$

where $Bio_x$ corresponds to the exported nutrient P or K [kg m$^{-2}$ a$^{-1}$] by harvest. $r_x$ is the P:N or K:N stoichiometric ratio used in Bodirsky et al. (2012). $N_{harvest}$ is the exported N for a minimum 0.7 kg m$^{-2}$ a$^{-1}$ or a maximum 3.6 kg m$^{-2}$ a$^{-1}$ harvest rate. The harvest rate value was based on the MAgPIE output for each grid cell, representing the minimum and maximum projected global harvest rate for a period of 95 years.

### 2.3. Geogenic P supply for AR

The geogenic P source databases have different spatial resolution (Table 2), we resampled each of them to a coarser 2°×2° spatial resolution fields by nearest neighbor interpolation to minimize distortions of location (Pontius, 2000). Nearest neighbor interpolation method reliably retain the overall proportions of an original fine resolution map (Christman and Rogan, 2012). As the uncertainty on which P pool is available for long-term plant nutrition is high (Johnson et al., 2003), two scenarios for soil P supply were investigated: scenario one considering P from weathering and atmospheric P deposition. Scenario two the same as scenario one plus inorganic labile P and organic P (Yang et al., 2014a).

The atmospheric dry and wet P deposition rates were taken from simulation outputs for the 2006 – 2013 period and for the years 2030, 2050, and 2099 for an RCP4.5 scenario for a grid cell size of 1° (Wang et al., 2017). The simulations were based on P emissions of sea salt, dust, biogenic aerosol particles, and P emitted by combustion processes, and performed by the global aerosol chemistry-climate model LMDZ-INCA (cf., Wang et al. (2017) for a detailed description of model and model assumptions). The simulation gaps were closed by linear regression and the cumulative atmospheric P deposition was calculated by summing up the deposition rate of each cell for 2006 – 2099 period according to Eq. (3):

$$P_{tot} = \sum_{i=2006}^{2099} P_i, \tag{3}$$

where $P_{tot}$ [kg m$^{-2}$] is the cumulative atmospheric P deposition of the 2006 – 2099 period (Fig. 3a). $P$ [kg m$^{-2}$ a$^{-1}$] is the
atmospheric P deposition of each year $i$ within a grid cell.

The total soil P map from Yang et al. (2014a) was used as estimation of the projected long term available P in the soil system
(Fig. 3b). The total P supply by weathering for the 21$^{th}$ century (2006 – 2099) was based on Hartmann et al. (2014) maps (Fig.
3c) that describes the chemical weathering as a function of runoff and lithology, being corrected to temperature and soil
thickness (Hartmann et al., 2014) and calibrated on 381 catchments in Japan (Hartmann et al., 2009). A relationship between
air temperature and weathering rate was used, which was derived from reconstructed weathering rates and different climate
change scenarios for the recent past (1860-2005) using the weathering model applied here. The relationship in which P
weathering increases by 9% per 1°C increase (Goll et al., 2014) implicitly accounts for changes in soil hydrology. Without
accounting for P concentration changes in primary and secondary P minerals. Due to the large uncertainties in projected
changes in soil hydrology we omitted a more detailed representation of hydrological effects on weathering.

## 2.4. Estimating geogenic P gap, related C-fixation reduction, and balanced Mg, Ca, and K supply for AR

The potential P gap ($P_{gap}$ [kg m$^{-2}$]) was estimated as the difference between additional mean and range (95$^{th}$ and 5$^{th}$ percentiles)
P demand estimated from the N stock for the two different AR scenarios (section on Afforestation/Reforestation nutrient
demand), and the geogenic P supply from the different supply scenarios ($P_{sup}$ [kg m$^{-2}$]) within the cover fraction for a grid cell
of biome i ($f_i$ [-]), for 21$^{st}$ century (2006 – 2099) according to Eq. (4):

$$P_{gap} = P_{sup} \times f_i - \Delta P_{pool,i}, \tag{4}$$

the plant C-fixation reduction was estimated based on the P gap and calculated following Eq. (5):

$$C = r_C \times P_{gap}, \tag{5}$$

where $C$ [kg m$^{-2}$] is the plant reduced C-fixation due to the projected P gap. $r_C$ is the used stoichiometric C:P ratio based on
mean and range (5$^{th}$ and 95$^{th}$ percentiles) chemistry for wood and leaves derived from the N-limited and N-unlimited AR
scenario N stock as described in subsection 2.2.1.

The Mg, Ca, and K necessary supply for balanced biomass nutrition ($M_x$ [kg m$^{-2}$]) should be proportional to the supplied P
($P_{EW}$ [kg m$^{-2}$]) and was calculated following Eq. (6):

$$M_x = r_x \times P_{EW}, \tag{6}$$

with $P_{EW}$ being equal to the projected $P_{gap}$ since it is covered by P from Enhanced Weathering according to Eq. (7):

$$P_{EW} = P_{gap}, \tag{7}$$

where $r_x$ is the used stoichiometric ratio Mg:P, Ca:P, K:P obtained by normalizing the N stock based additional Mg, Ca, and K demand to the N stock based additional P demand.

### 2.5. Enhanced Weathering Mg, K, Ca, and P potential supply

To cover the potential of different igneous rocks for EW strategies, rhyolite and dacite (acidic rocks), andesite (intermediate rock) and basalt (basic rock) were selected to project necessary amounts to cover P gaps from the AR scenarios. Data on macronutrient concentrations (Mg, Ca, K, P) in weight percent within these rocks were downloaded from the Earthchem web portal (Fig. 4; http://www.earthchem.org, accessed on 2017-07-14). The data was selected for rocks named as rhyolite, dacite, andesite, and basalt. Neglecting intermediate compositions between different lithotypes (i.e., a trachybasalt that has its chemical composition lying in between trachyte and basalt). Rocks that were under any metamorphism grade (e.g., meta-basalt) were neglected because metamorphism can change rock mineralogy. We neglected rocks known to have high content of minerals rich in trace elements (e.g., an alkali basalt can have P concentration >3000 ppm (Porder and Ramachandran, 2013), but it is rich in olivine (John, 2001;Irvine and Baragar, 1971) that contains elevated concentrations of nickel and chromium (Edwards et al., 2017)). Nickel and chromium are trace elements problematic for agriculture (Edwards et al., 2017). Thus, following the classification criteria, the number of selected data to calculate descriptive statistics for Mg, Ca, K, P content within rocks were 2985 chemical analysis for rhyolite, 3008 chemical analysis for dacite, 11099 chemical analysis for andesite, and 23816 chemical analysis for basalt.

The nutrient supply was estimated assuming complete rock powder dissolution in the system considering median and ranges ($5^{th}$ or $95^{th}$ percentile) chemical composition. The duration of complete rock powder dissolution varies depending on the grain size (i.e., one year for grain sizes between 0.6 – 90 µm (Strefler et al., 2018) for basalt). The results and discussion will focus on basalt rock powder considering median P values (500 ppm) and range ($5^{th}$ (157 ppm) and $95^{th}$ (1833 ppm) percentiles), as basalt is abundant worldwide (Amiotte Suchet et al., 2003;Börker et al., 2019) and has a high P content compared to acidic and intermediate rocks (Porder and Ramachandran, 2013). Median P concentration can be >3000 ppm for alkali basalts, but for a broader basalt classification that considered 97895 samples, it can be of 916 ppm (Porder and Ramachandran, 2013). The necessary mass of rock powder to supply macronutrients (Mg, Ca, K, or P) was calculated following Eq. (8):

$$R_d = \frac{M_{ex}}{f_{nut}}, \tag{8}$$

where $R_d$ [kg rock m$^{-2}$ or kg rock m$^{-2}$ a$^{-1}$] represents the mass of a rock type to cover AR or BG nutritional needs, $M_{ex}$ [kg m$^{-2}$ or kg m$^{-2}$ a$^{-1}$] is the mass of required nutrient for AR or BG (e.g., P to cover a $P_{gap}$ obtained by Eq. (4)), and $f_{nut}$ [-] is the median and range ($5^{th}$ or $95^{th}$ percentile) fractions of interest nutrient within the selected rock.

However, the potential nutrient supply by EW for different amounts of rock powder being deployed was also estimated following Eq. (9):

$$Nut_{in} = M_{rock} \times f_{nut}, \tag{9}$$

where $Nut_{in}$ [kg m$^{-2}$ or kg m$^{-2}$ a$^{-1}$] represents the macronutrient input by dissolving a chosen rock. $M_{rock}$ [kg rock m$^{-2}$ or kg rock m$^{-2}$ a$^{-1}$] is the mass of rock added to the natural system.

## 2.6. Related impacts on soil hydrology from Enhanced Weathering deployment

Large scale deployment of rock powder on soils is expected to influence its texture. The deployed amount and texture of rock powder will somehow affect hydraulic conductivity, water retention capacity, and specific soil surface area. Pedotransfer functions (PTFs) are used to estimate soil hydraulic properties (Schaap et al., 2001;Whitfield and Reid, 2013;Wösten et al., 2001) and such approximations have proven to be a suitable approach (Vienken and Dietrich, 2011). PTFs make use of statistical analysis (Saxton and Rawls, 2006;Wösten et al., 2001), artificial neural networks, and other methods applied to large soil databases of measured data (Wösten et al., 2001). The equations from Saxton et al. (1986) performed the best estimations of soil hydraulic properties (Gijsman et al., 2002). Later on, Saxton and Rawls (2006) improved Saxton et al. (1986) PTFs and they are used to estimate the effects on soil hydraulic properties due to deployment of basalt powder (Eqs. (10) – (18)).

The potential changes in soil hydraulic properties, due to the application of a fine basalt texture (15.6% clay, 83.8% silt, and 0.6% fine sand) or a coarse basalt texture (15.6% clay, 53.8% silt, and 30.6% fine sand) were estimated as a function of rock powder deployment for soils corresponding to P gap areas from the N-unlimited AR scenario. According to the international organization for standardization, the man-made materials can be classified according to their grain sizes; therefore, here the clay comprehends grain diameters $\leq 2$ µm, silt comprehends grain diameter $2 - 63$ µm, and fine sand comprehends $63 - 200$ µm (ISO 14688-1:2002), but since full dissolution is assumed, the ground basalt fine sand encompass grain sizes of diameter $63 - 90$ µm remaining withing the ISO 14688-1:2002 classification. The N-unlimited AR scenario was selected since it would have the highest P deficiencies requiring more rock powder to cover the P gaps (i.e., it represents the maximum effect). The estimations are for a homogeneous mixture of rock powder and topsoil depth of 0.3 m. Downward transport of fine-grained material is neglected for simplification. The considered values represent upper limits of rock powder application. The impacts on plant available water (PAW) is given by the difference between water content at a pressure head of -33 kPa (Eq. (11)) and -1500 kPa (Eq. (10)), while the impact on soil hydraulic conductivity is given by (Eq. (14); Saxton and Rawls, 2006):

$$\theta_{1500} = \theta_{1500t} + (0.14 \times \theta_{1500t} - 0.02), \tag{10}$$

$$\theta_{33} = \theta_{33t} + (1.283 \times (\theta_{33t})^2 - 0.374 \times (\theta_{33t}) - 0.015), \tag{11}$$

$$\theta_{(S-33)} = \theta_{(S-33)t} + (0.636 \times \theta_{(S-33)t} - 0.107), \tag{12}$$

$$\theta_S = \theta_{33} + \theta_{(S-33)} - 0.097 \times S + 0.043, \tag{13}$$

$$K_S = 1930 \times (\theta_S - \theta_{33})^{(3-\lambda)}, \tag{14}$$

with:

$$\theta_{1500t} = -0.024 \times S + 0.487 \times C + 0.006 \times OM + 0.005 \times (S \times OM) - 0.013(C \times OM) + 0.068(S \times C) + 0.031, \tag{15}$$

$$\theta_{33t} = -0.251 \times S + 0.195 \times C + 0.011 \times OM + 0.006 \times (S \times OM) - 0.027 \times (C \times OM) + 0.452(S \times C) + 0.299, \tag{16}$$

$$\theta_{(S-33)t} = 0.278 \times S + 0.034 \times C + 0.022 \times OM - 0.018 \times (S \times OM) - 0.027 \times (C \times OM) - 0.584 \times (S \times C) + 0.078, \tag{17}$$

$$\lambda = \left[\frac{\ln(1500) - \ln(33)}{\ln(\theta_{33}) - \ln(\theta_{1500})}\right]^{-1}, \tag{18}$$

where $S$ and $C$ respectively represent the soil texture corresponding to sand and clay diameters [wt %], $OM$ is the soil organic matter [wt %], the moisture [wt %] are estimated by $\theta_{1500}$ and $\theta_{33}$ respectively representing the soil moisture for a pressure head of -1500 kPa ($R^2 = 0.86$) and of -33 kPa ($R^2 = 0.63$). $\theta_{(S-33)}$ and $\theta_S$ respectively corresponds to the 0 kPa to -33 kPa moisture ($R^2 = 0.36$), and to the saturated (0 kPa) moisture ($R^2 < 0.25$). $K_S$ [mm h$^{-1}$] represents the saturated soil hydraulic

conductivity and $\lambda$ is the slope of the logarithmic tension-moisture curve. The numbers in front of each described variable are regression coefficients (Saxton and Rawls, 2006).

The initial hydrologic properties of topsoil were estimated for a depth of 0.3 m, as it is the average depth usual machinery can homogeneously mix topsoil (Fageria and Baligar, 2008). Greater depths can be reached but under higher energy and labor costs (Fageria and Baligar, 2008). The global data set of derived soil properties (Batjes, 2005), which had textural information

(sand, silt, and clay content) for shallow soil depths (0.3 m) was used. The raster had a resolution of 0.5° and the soil properties for the interest areas of biomass growth limitation (Supplement S1 Fig. S. 7a) were included by a spatial join (using Esri ArcMap 10.8). The nutrient deficient areas encompass soils of different textures and organic matter content, which had their initial $K_S$ estimated separately based on Eq. (14). The sum of clay, silt, and sand fractions within each cell should always be unity, and were corrected when necessary by Eq. (19):

$$G_{cor} = \frac{(G_{ini} \times M_{soil\_cell})}{\sum(G_{ini} \times M_{soil\_cell})}, \tag{19}$$

with:

$$M_{soil\_cell} = V_{cell} \times \rho_{bulk\_cell}, \tag{20}$$

where $G_{ini}$ represents the initial topsoil texture (sand, silt, and clay content) of a specific raster cell [-]. $V_{cell}$ [km$^3$] is the raster cell volume obtained by multiplying the area [km$^2$] to the soil depth of $0.3\times10^{-3}$ km. $\rho_{bulk\_cell}$ [kg km$^{-3}$] is the raster cell topsoil bulk density. $M_{soil\_cell}$ [kg] is the total soil mass of a raster cell. $G_{cor}$ [-] is the corrected soil texture (sand, silt, and clay content).

The necessary rock powder mass was estimated by Eq. (8) to close the $P_{gap}$ obtained by Eq. (4). The effect of basalt powder application in soil $K_S$ and PAW was estimated by assuming a homogeneous mixture between applied basalt powder and topsoil. The changes on initial soil organic matter (SOM) concentration within a raster grid-cell were obtained by normalizing the SOM to the sum of applied basalt mass, mass of soil, and initial SOM mass by Eq. (21). This was necessary since the SOM concentration at the moment of basalt deployment would have a relative decrease compared to initial SOM concentration:

$$OM_c = \frac{OM_{cell}}{M_{b\_cell} + M_{soil\_cell} + OM_{cell}} \times 100, \tag{21}$$

with:

$$OM_{cell} = OM_{wt\%} \times M_{soil\_cell}, \tag{22}$$

where $OM_c$ [wt %] is the corrected soil organic matter content, $OM_{cell}$ [kg] is the organic matter mass within the raster cell. $M_{b\_cell}$ and $M_{soil\_cell}$, both in [kg], are the mass of basalt and mass of soil for a specific raster cell.

The impacts in soil texture by rock powder application considered the textures of applied basalt mass added to the initial soil mass by Eq. (23). It was assumed a content of 15.6% clay, 83.8% silt, and 0.6% fine sand for fine basalt powder and 15.6%
clay, 53.8% silt, and 30.6% fine sand for a coarse basalt powder.

$$G_{bs} = \frac{\left(G_{ini} \times M_{sed_{cell}} + M_{b_{cell}} \times G_{basalt}\right)}{\Sigma\left(G_{ini} \times M_{sed_{cell}} + M_{b_{cell}} \times G_{basalt}\right)}, \tag{23}$$

where $G_{basalt}$ corresponds to the texture fractions of the fine or coarse basalt powder. $G_{bs}$ corresponds to the texture fractions of resulting mixture of basalt plus soil. Thus, the texture fractions of resulting mixture of basalt plus soil obtained by Eq. (23) were replaced within Eq. (15) to Eq. (17) to estimate the impacts in soil hydraulic conductivity by Eq. (14) and PAW by subtracting the outcome from Eq. (11) to the outcome from Eq. (10), with clay size (grains >1 µm and <3.9 µm) being the
finest grain size we can consider.

Besides texture and organic matter, intrinsic grain properties (e.g., the shape of grains and pores, tortuosity, specific surface area, and porosity) should be considered (Bear, 1972). The equations from Beyer (1964) are based on the non-uniformity of grain size distribution and density of the grain packing to estimate soil properties. Carrier III (2003) uses information on the particle grain size distribution, the particle shape, and the void ratio on his equations to estimate soil properties. However, such

 detailed information on a global scale is missing turning Beyer (1964) and Carrier III (2003) equations not applicable to our analysis.

## 3. Results

### 3.1. Afforestation/Reforestation P gaps and Enhanced Weathering as nutrient source

The global C sequestration for the N-limited AR scenario is 190 Gt C, while for the N-unlimited AR scenario it is 34 Gt C higher. The AR model from Kracher (2017) shows an increase in biomass production in tropical and temperate zones (Fig. 2). The results only focus on the N-limited scenario since it considered natural N supply, but the results for the N-unlimited scenario are presented only in the supplement (Supplement S1 section B ii). The calculated P budgets according to Eq. (4) for the AR time of 2006 – 2099 (Fig. 5) considered different geogenic supply scenarios (scenario one: P from weathering and atmospheric P deposition; scenario two: the same as scenario one plus inorganic labile P and organic P) and the average and range N stock-based P demand (calculated following Eq. 1) for the AR simulation from Kracher (2017).

The ideal P biomass additional demand (calculated from Eq. (1)), to sequester 190 Gt C (N-limited AR scenario) amounts to 200 Mt P on global scale for a mean wood and leaves P content; for 5th and 95th percentile, the estimated P demand would be 71 and 345 Mt P respectively. The P budget (estimated from Eq. (4)) for geogenic P supply scenario one suggest that P deficiency areas are distributed around the world, but with more frequent occurrences in the northern hemisphere (Fig. 5a) and the P gaps can potentially reach up to ~17 g P m$^{-2}$ (~4 – ~30 g P m$^{-2}$ for 5th and 95th quartiles of wood and leaves chemistry; Table 3) or a global P gap of ~77 Mt P (~9 – 181 Mt P$^2$ for 5th and 95th quartiles of wood and leaves chemistry; Table 3). However, for geogenic P supply scenario two, the P deficiency areas are predominantly located in the southern hemisphere (Fig. 5c) and the P gaps can potentially reach up to ~7 g P m$^{-2}$ (~2 – ~12 g P m$^{-2}$ for 5th and 95th quartiles of wood and leaves chemistry; Table 3) or a global P gap of ~10 Mt P (1 – ~35 Mt P$^2$ for 5th and 95th quartiles of wood and leaves chemistry; Table 3).

The P and N limitation cause an average C reduction of 47% for the geogenic P supply scenario one and 19% for the geogenic P supply scenario two (obtained by accounting the C reduction from N limitation, which is 34 Gt C plus the C reduction from Table 3 and then normalized by the global sequestration for the N-unlimited scenario of 224 Gt C) or ~-1.1 and ~-0.5 Gt C a$^{-1}$, respectively. In some areas, the C sequestration can be reduced by up to 100% compared to the predicted C sequestration of the AR models (Fig. 6). Accounting for N and P limitation on AR suggests that the biomass production will be affected, consequently decreasing the C sequestration potential of AR strategies (Table 3 and Fig. 6). Therefore, supplying the demanded P would positively contribute to biomass to reach the predicted growth of the specific AR scenario.

Besides removing carbon from the atmosphere, EW can also amend soils by supplying nutrients and increasing alkalinity fluxes (Leonardos et al., 1987;Nkouathio et al., 2008;Beerling et al., 2018;Hartmann et al., 2013;Anda et al., 2015). Since

basalt has higher P content compared to acidic and intermediate rocks (Porder and Ramachandran, 2013), it could be used as raw material for EW to cover the estimated P gaps of Fig. 5a and Fig. 5c. For a median Basalt P content of 500 ppm (cf., subchapter 2.5), it would be necessary to apply ~33 and ~13 kg basalt $m^{-2}$ (Fig. 5b and Fig. 5d) in areas of high P deficiency (~17 and ~7 g P $m^{-2}$, Fig. 5a and Fig. 5c respectively), considering the AR time span, the deployment rates would be less than 1 kg basal $m^{-2}$ $a^{-1}$, if full congruent dissolution occurs as assumed for further given scenarios.

The total amount of basalt powder to close the estimated P gaps from Fig. 5 would depend on the assumed geogenic P supply scenario and chemical composition of wood and leaves, but for a mean P chemical composition, at least ~153 Gt basalt would be necessary for geogenic P supply scenario one and ~20 Gt basalt for geogenic P supply scenario two. Basalt has a carbon capture potential of ~0.3 t $CO_2$ $t^{-1}$ basalt (Renforth, 2012), resulting in ~46 Gt $CO_2$ (~12.4 Gt C) and 6 Gt $CO_2$ (1.6 Gt C) capture by closing the P gaps from Fig. 5a and Fig. 5c, respectively. If wood and leaves P concentration correspond to 5th percentiles (Table 1) ~2 Gt basalt would be needed for closing the P gaps from a geogenic P supply scenario two (Supplement S1 Fig. S1), which would potentially sequester ~0.6 Gt $CO_2$ (~0.2 Gt C) due to weathering. If wood and leaves P concentration correspond to 95th percentiles (Table 1) ~362 Gt basalt for closing the P gaps from a geogenic P supply scenario one (Supplement S1 Fig. S3) would be necessary, which would potentially sequester ~98 Gt $CO_2$ (~27 Gt C) due to weathering. The amount of basalt needed was estimated for a P content of 500 ppm and an increase in basalt P concentrations would represent a decrease in the necessary amounts of basalt powder. Incongruent dissolution of basalt might occur consequently increasing the necessary amounts of deployed basalt to cover the estimated P gaps.

Basalt deployment can also guarantee a balanced supply of Mg, Ca, and K for different deployment rates (Fig. 7), potentially preventing the shift of growth limitation to some of these nutrients within the P gapped areas (Fig. 5). Besides basalt, rhyolite, dacite or andesite could alternatively be used as a source of P, but these rocks generally have lower P content (Fig. 4). As a consequence, the necessary amounts of rhyolite, dacite or andesite would be higher than that for basalt. Even though, for a median rock nutrient content, if these rocks are used to close the projected P gaps, they potentially can supply the necessary amount of Ca, Mg, and K for balanced tree nutrition (Fig. 8).

### 3.2. Enhanced Weathering coupled to bio-energy grass production

For the simulation time spam of 1995 – 2090 the minimum and maximum biomass growth yield amounts to 0.7 and 3.6 kg $m^{-2}$ $a^{-1}$, which represent a K export of 4.2 – 22 g $m^{-2}$ and a P export of 0.7 – 3.6 g $m^{-2}$ according to Eq. (2). To guarantee maximum bioenergy grass yield, the exported nutrients should be replaced. For a high nutrient content (95th quartile) deploying up to 1.5 kg basalt $m^{-2}$ $a^{-1}$ could meet the K needs of bio-energy grass (Fig. 9) and would be able to replenish up to 75% of the exported P, if the maximum bio-energy grass yield is considered (Fig. 9). Industrial fertilizer co-application would be indicated to completely replenish exported P reducing industrial fertilizers dependency. Deploying 8 kg basalt $m^{-2}$ $a^{-1}$ would be enough to replenish exported K and P by harvest assuming median nutrient content of basalt powder, congruent and complete dissolution (Fig. 9).

### 3.3.    Impacts on soil hydrology

The baseline hydraulic properties for soils within the P gap areas from the N-unlimited AR scenario, since this scenario represents the maximum effect, were estimated by Eq. (10), and they show high variability. The projected hydraulic conductivity ($K_S$) of top soils for areas corresponding to those of P budget from geogenic P supply scenario one (Supplement S1 Fig. S7a), for the N-unlimited AR scenario encompass values ranging from $1.5 \times 10^{-7}$ and $7.8 \times 10^{-5}$ m s$^{-1}$ and for PAW of 4% and 32% (Table 4). Neglecting the topography, soils having low $K_S$, (e.g., values of $1.5 \times 10^{-7}$ m s$^{-1}$) would experience the lowest water infiltration rate. The impacts of deploying a fine basalt texture (15.6% clay, 83.8% silt, and 0.6% fine sand) or a coarse basalt texture (15.6% clay, 53.8% silt, and 30.6% fine sand), which are in the range of commercial powders (Nunes et al., 2014), on soil hydrology were estimated by Eq. (10) for different application upper limits.

The effects of rock-powder deployment could be neglected, on average, for upper limits of 50 kg basalt m$^{-2}$ for a fine and 205 kg basalt m$^{-2}$ for a coarse textured rock-powder. However, deviations from what is expected for the mean might occur (Fig. 10 and Fig. 11). The average values for PAW increase together with the increase of the upper limits of rock powder application, but for a coarse basalt powder some areas might experience a decrease in the PAW (Fig. 10 and Fig. 11).

Closing the observed P gap areas from the N-unlimited AR scenario would require a maximum deployment of 34 kg basalt m$^{-2}$ if geogenic P supply scenario one is assumed and 13 kg basalt m$^{-2}$ if geogenic P supply scenario two is assumed (Supplement S1 Fig. S7). Filling the P gaps from scenario two by a coarse or fine basalt powder (given complete dissolution of P-bearing minerals) the related changes in soil hydrology would remain below ±10% for most of the areas (Supplement S1 Fig. S12). If the geogenic P supply from scenario one, for the N-unlimited AR scenario (Supplement S1 Fig. S7a), is assumed and a fine basalt powder is applied, the changes on hydraulic conductivity range between 58% and -11% (Fig. 12a). Decrease on PAW could be neglected for most of the deployment areas, but some would have an increase of up to 31% from 13.8% to 18.2% (Fig. 12c). A coarse basalt powder would, in general, cause fewer impacts to soil hydraulic properties (Fig. 12b and Fig. 12d).

## 4.    Discussion and implications

### 4.1.    Enhanced Weathering coupled to Afforestation/Reforestation

Phosphorus (P) is a limiting nutrient in a wide range of ecosystems (Elser et al., 2007) and in temperate and tropical climate zones (Du et al., 2020). P deficiency might affect biomass growth of tropical (Herbert and Fownes, 1995;Tanner et al., 1998;Wright et al., 2011) and northern forests (Menge et al., 2012;Shinjini et al., 2018) with mineral P already limiting biomass production in European forests (Jonard et al., 2015) and in Forests from USA (Garcia et al., 2018), as well as agricultural areas (Ringeval et al., 2019;Kvakić et al., 2018). The uncertainty on which P pool is available for long-term plant nutrition is high (Johnson et al., 2003;Sun et al., 2017) and we tackled this uncertainty assuming two potential geogenic P supply scenarios. Geogenic supply scenario two, assuming P from weathering and atmospheric deposition plus inorganic labile P and organic P,

is a very optimistic assumption that might not correspond to reality based on the already observed P limitation on different ecosystems (Elser et al., 2007). However, we cannot rule out that gradual shifts in soil organic P fractions occur, which make comparable amounts of P as in scenario two available over time.

The numerical simulations of Kracher (2017) predicted biomass growth for the 21st century (Fig. 2) considering natural water supply, $CO_2$ fertilization, and N-unlimited and N-limited scenario for an RCP4.5 greenhouse gas concentration trajectory and land use transitions. The predicted C sequestration by the N-limited AR scenarios from Kracher (2017) is ~2 Gt C $a^{-1}$. Different authors reported the potential C sequestration by afforestation or reforestation being of 0.3 – 3.3 Gt C $a^{-1}$ for the end of 2099 (National Research Council, 2015;Lenton, 2014, 2010;Smith et al., 2015 apud Fuss et al., 2018). However, the predicted sequestration potential estimated by Kracher (2017) can drop to ~1.3 Gt C $a^{-1}$ if geogenic P supply scenario one for mean P content within wood and leaves is selected. If geogenic P supply scenario two for mean P content within wood and leaves is selected, it drops to ~1.9 Gt C $a^{-1}$.

More than 60,000 tree species are recorded worldwide (Beech et al., 2017) and a precise estimation on tree chemistry represents a challenge, which we attempted to represent by the considered ranges of wood and leaves chemistry from the databases. However, different pathways and mechanisms control soil P availability to the plant (Vitousek et al., 2010), and they are not considered in our estimations leading to conservative predictions. Adding soil P dynamics to models would allow to reliably quantify the C sequestration potential of AR (e.g., using P enabled land surface models; Sun et al., 2017;Wang et al., 2017;Goll et al., 2012;Goll et al., 2017;Wang et al., 2010;Yang et al., 2014b).

Kracher (2017) has shown that N can limit biomass production and consequently C sequestration. To achieve the projected C sequestration of 190 Gt C for N-limited scenario, the estimated P gaps must be closed. Potential P sources are industrial fertilizers, like diammonium phosphate (DAP) or rock powder (e.g., basalt). However, DAP potentially represents an extra input of ammonium to the groundwater and it is expected, in the long-term, that DAP deployment acidifies the soil (McLaughlin, 2016).

Most of the world soils are acidic, with some being strongly acidic (IGBP-DIS, 1998), which generally favors the sorption of orthophosphate onto Fe- and Al-(hydro)oxides surfaces and clay minerals, essentially demobilizing P (Shen et al., 2011). Besides that, the long AR time span can undermine the effectiveness of DAP to supply P for forests due to the high soil acidification potential of DAP. Therefore, rock powder application can be an alternative as nutrients are slowly released and an increase of alkalinity fluxes is expected (Dietzen et al., 2018), which can raise and stabilize the pH of soils.

Re-establishing soil pH to (near) neutral conditions, generally between 6.6 and 7, will provide new nutrient holding sites at Fe- and Al-(hydro)oxides surfaces, and at soil organic matter; which turns the sorbed orthophosphate plant available. An application of 8 kg $m^{-2}$ basalt powder can increase the CEC of oxisols by 150 – 300% (Anda et al., 2015;Anda et al., 2009) and improve the C- and N-mineralization (Mersi et al., 1992), for ultisols, the CEC increase by 44% after deploying ~7 kg $m^{-2}$ basalt powder (Noordin et al., 2017).

To avoid shifts of nutrient limitation, the supply of macronutrients like Mg, Ca, and K might be proportional to P supply since Mg is required as an essential element in chlorophyll, Ca has a structural role, and K is responsible for water and ionic balance (Hopkins and Hüner, 2008). Rock powder can be used as source of these nutrients, as suggested by different authors (Beerling et al., 2018;Hartmann et al., 2013;Straaten, 2007) and according to our results from Fig. 7 and Fig. 8. However, the potential of basalt powder to supply K, based on chemical composition, is lower than for other analyzed rocks. For median values, rhyolite has the highest content of K; however, if occurring in K-feldspars it will not be plant available. Blending these rocks in different proportions could result in a more balanced macronutrient supply (Leonardos et al., 1987).

RCP8.5 scenario predicts that global agricultural areas (crop land and pastures) are going to increase in the course of $21^{st}$ century due to a decrease in forested area (Sonntag et al., 2016). Assuming a future scenario of high atmospheric $CO_2$ levels (RCP8.5), but using the land use transitions and wood harvest rates from a RCP4.5 scenario (Sonntag et al., 2016), a similar forest cover fraction than the one presented in Fig. 2 is expected (cp. Figure 1 Sonntag et al. (2016)) and geogenic P supply would also limit the predicted biomass growth. Similar areas of forest growth were observed in Figure 2c presented in the study from Yousefpour et al. (2019) by comparing it to Fig. 2. Though using only one model induces uncertainty, however, it would not change the general message of this work.

## 4.2.    Enhanced Weathering coupled to bio-energy grass production

Generally, natural soil P content is inadequate for long-term cultivation of agricultural plants. To overcome this issue, P is supplied by fertilizers to reach or maintain optimum levels of crop productivity (Sharpley, 2000) after several harvest rotations. In order to keep a positive $CO_2$ balance, an alternative to industrial fertilizers might be used to replenish the exported nutrients by harvest. The chemical composition of rocks is highly variable (Fig. 4) and different rock types can be used for EW. Ideal rock types need to be chosen in order to resolve a specific plant nutrient deficiency, and enhance the nutrient reservoir of a target soil besides increasing the soil pH, the CEC (Anda et al., 2015;Anda et al., 2009), improve the C- and N-mineralization (Mersi et al., 1992), the soil organic carbon (Doetterl et al., 2018) and the supply of Si (Beerling et al., 2018;Hartmann et al., 2013). In the case of oxisols, which are found over about 8% of the glacier-free land surface and common in tropical and subtropical agricultural regions, application of 8 kg m$^{-2}$ basalt powder can increase the CEC by 150 – 300% (Anda et al., 2015;Anda et al., 2009). For ultisols, which are found over about 8% of the glacier-free land surface, application of ~7 kg m$^{-2}$ basalt powder can increase the CEC by 44% (Noordin et al., 2017).

Overall, rock application has the potential to resupply the harvest exported nutrients, and partially or totally close the short- and long-term nutrient gaps in soil. Individual rock types, from basic (Mg, Ca) to acidic (K, Na), contain varying amounts of target nutrients and mixing them might increase the overall nutrient supply capacity (Leonardos et al., 1987). Intrinsic mineralogical and or petrographic structures can influence the release of nutrients (Ciceri et al., 2017), which makes them plant unavailable in some cases. K can also limit plant growth; it occurs in K-feldspars as a plant unavailable form, in the case of acidic rocks, but becomes accessible after hydrothermal treatment (Liu et al., 2015;Ma et al., 2016a;Ma et al., 2016b).

However, research on release processes of other macro- and micronutrients and on nutrient-release optimization (e.g., by hydrothermal decomposition) is necessary to be able to parameterize this effect in the soil environment.

Harvest rates will control the nutrient export from bioenergy grass fields. Therefore, an increase in harvest rate represent an increase in nutrient export and vice-versa. Thus, to keep with a sustainable nutritional balance of soils, the exported nutrients must be replenished, otherwise maintaining the high harvest rates become an unsustainable situation. Accounting for other simulation setup or a numerical model different from MAgPIE might change the harvest rates of this study. If we assume that the maximum harvest rate of 3.6 kg m$^{-2}$ a$^{-1}$ would hypothetically increase by one order of magnitude, the maximum exported nutrients would be of ~0.2 kg K m$^{-2}$ a$^{-1}$ and ~0.04 kg P m$^{-2}$ a$^{-1}$, which would demand a basalt deployment rate of ~13 kg m$^{-2}$ a$^{-1}$ and ~20 kg m$^{-2}$ a$^{-1}$ (considering 95$^{th}$ percentiles of chemical composition for basalt) to respectively replenish the exported nutrients. If median K and P concentrations on basalt powder are assumed, the basalt deployment rate increase to ~48 kg m$^{-2}$ a$^{-1}$ and 73 kg m$^{-2}$ a$^{-1}$ to respectively replenish the exported nutrients (Supplement SI Fig. S11). However, such an increase in harvest rates might not correspond to reality. Harvest rates smaller than 0.7 kg m$^{-2}$ a$^{-1}$ (the minimum) represent less nutrient export, decreasing the basalt powder deployment rates necessary to replenish the exported nutrients by harvest.

### 4.3.    Impacts on soil hydrology

AR and BECCS demand huge quantities of irrigation water (Boysen et al., 2017b;Bonsch et al., 2016), and it is projected that climate change will affect the water balance, and consequently influence crop yields (Kang et al., 2009). Soils with higher water holding capacity will better tolerate the impacts of drought (Kang et al., 2009). Therefore, practices that improve water availability to plants at the root system are used as strategies to mitigate drought effects (Rossato et al., 2017). We investigated if deployment of rock powder can change the top soil hydraulic conductivity, and plant available water (PAW) for different application ranges.

Concrete effects of EW on biomass productivity would depend if the changes in the initial PAW values for top soils would reach PAW threshold values to trigger biomass productivity (Sadras and Milroy, 1996). In general, the average changes on topsoil PAW related to basalt powder application would not be enough to trigger biomass growth. Therefore, areas showing PAW changes from 14% to 21% would not trigger leaf and stem expansion of maize, wheat or soybean (Sadras and Milroy, 1996), but could increase leaf and stem expansion of pearl millet (Sadras and Milroy, 1996) after deploying 50 kg basalt m$^{-2}$ with a fine texture. 50 kg basalt m$^{-2}$ of coarse powder changes PAW by 19% consequently not triggering biomass productivity. The finest grain size Saxton and Rawls (2006) equations can consider is the clay fraction (grains diameter >1 μm and <3.9 μm). Fine grain sizes influence the exposed reactive surface area of rock powder, which will affect the weathering rates. The fine basalt would have the grain sizes ranging in between 0.6 – 90 μm which might be enough to completely dissolve the deployed rock powder after one year (Strefler et al., 2018). For the coarse basalt powder, ~70% of its granulometry fall into the 0.6 – 90 μm range and from the other 30%, about 20% might be dissolved in one year (Strefler et al., 2018). Based on the used pedotransfer functions, If a basalt powder would contain only grains on the clay size fraction, the effects on soil hydraulic

conductivity would decrease by 37% for deployment amount of 30 kg basalt $m^{-2}$ (for the fine rock powder used in our work, the hydraulic conductivity would decrease by only 2%). The finer the grain gets the higher the energy input for grinding is, which can drastically affect the costs of EW (it can reach up to 500\$ $tCO_2^{-1}$ sequestered; Strefler et al., 2018). Since grains of different diameters need different times for complete dissolution, a rock powder with different grain sizes would act as a constant source of nutrients to soil.

During the weathering of rock powder, clay mineral genesis can occur and potentially increase the water holding capacity of soils (Gaiser et al., 2000), which can subsequently change the estimated PAW. The added fresh silicate minerals to the soil by EW will have high reactivity releasing a significant amount of nutrients, which increases soil nutrient pools. The increased nutrient availability will increase the potential of soils to stabilize carbon (Doetterl et al., 2018) and a positive effect on PAW is expected to occur based on Eqs. (15) to (17) and according to Olness and Archer (2005). The suitable amounts of rock powder applied depend on the target changes of the chosen soil, and on its intrinsic grain size distribution and organic matter content. Intrinsic grain properties like the shape of grains and pores, tortuosity, specific surface area, and porosity should be considered (Bear, 1972) for the evaluation of changes in soil hydraulic properties by pedotransfer functions and its consequences to dissolution kinetics. A large set of data from field and laboratory experiments covering different soil types, climatic regions, and plant species would enable a qualitatively, and quantitatively reliable assessment of soil hydrology impacts, but also dissolution rates and changes on soil's mineralogy. The effects on soil microorganisms should be taken into account in order to correct the limits of rock powder deployment. The potential of rock powder to trigger plant suffocation, if gas exchange is prevented by water saturation of pores (Sairam, 2011), should also be considered before deployment.

### 4.4.    Challenges of rock powder deployment

Average tillage depth is 0.3 m and greater depths can be reached with  higher energy and labor costs (Fageria and Baligar, 2008). Since annual crops have an effective rooting depth typically in the range of 0.4 – 0.7 m (Madsen, 1985;Aslyng, 1976;Munkholm et al., 2003;Olsen, 1958), a deployment depth of 0.3 m seems to be reasonable.

Since tillage can trigger soil carbon loss (Reicosky, 1997;La Scala et al., 2006), deploying rock powder at soil surface might be a solution. At the soil surface, the long-term water percolation, and/or bioturbation (Fishkis et al., 2010;Taylor et al., 2015) can transport and mix fine-grained material to deeper regions within the soil profile, which potentially can change the $K_S$, and PAW at crop rooting zones. Groundwater recharge rates might change if clogging of pores at deeper regions of soil profile occurs or if the changes in soil hydraulic properties due to rock deployment can significantly influence the initial soil hydraulic conditions for a constant water precipitation. Taylor et al. (2015) argue that downward transport of a silt-textured powder deployed at a soil surface would easily reach the rooting zone of trees, which is in its majority in a depth up to 0.4 m. The authors suggest that in tropical regions higher depths might be reached due to intensive rain and bioturbation.

Detailed field studies to better comprehend downward transport of grained material through the soil profile, changes on soil water residence time, PAW, mineralogy, nutrient pools, CEC (Anda et al., 2015, 2013), and bioavailability of released trace metals (Renforth et al., 2015) are necessary. This would provide management recommendations for the diverse existing settings for EW application. In the present study, estimates for different basalt powder application upper limits are done for changes in soil hydraulic properties without accounting for downward transport of fine particles through the soil profile.

Besides avoiding clogging of pores of the top soil layer by rock powder application in a certain extent, downward transport of rock powder can contribute freshly ground material being in contact with roots of trees or crops, which can enhance the weathering rates and create new sites to retain nutrients (Kantola et al., 2017;Anda et al., 2015).

Once the freshly ground material is in contact with the soil, different factors control the nutrient supply efficiency of rock powder. The nutrients from fresh material are initially inert protected within the crystallographic structures of the minerals, and would become plant-available only in solution or associated to mineral surfaces (Appelo and Postma, 2005). The release of nutrients by weathering is controlled by film and intra-particle diffusion-limited mass transfer influenced by pH, and ionic strength of the soil aqueous solution (Grathwohl, 2014), both being controlled by rooting exudates in the rhizosphere and chemical composition of infiltrating waters.

Full dissolution is a simplification based on modelled scenarios (Taylor et al., 2015;Strefler et al., 2018). Under field conditions, soil water could rapidly reach near-equilibrium concentrations (Grathwohl, 2014), which would decrease weathering rates. The opposite would occur if near-equilibrium conditions could be disturbed by a sink of nutrients by nutrient root uptake (Stefánsson et al., 2001) or by percolation of water un-equilibrated with soil porous water (Calabrese et al., 2017). The nutrient (Mg, Ca, K, P, etc.) content of rocks can vary significantly. Besides that, deploying rock powders with grain sizes > 90 µm would decrease the reactive surface area of deployed rock powder decreasing the weathering fluxes (Goddéris et al., 2006). The median and the ranges ($5^{th}$ or $95^{th}$ percentile) values for Mg, Ca, K, P content obtained from the EarthChem database considered chemical analysis of 2985 rhyolites, 3008 dacites, 11099 andesites, and 23816 basalts. Broadening the classification criteria for these rocks would change median and the ranges ($5^{th}$ or $95^{th}$ percentile) for chemical composition; however, the selected median and the ranges of this study are conservative estimates. As an illustration, Porder and Ramachandran (2013) adopted another selection criteria, for the same database, which resulted in a total of 97895 samples and estimated a median P content of Basalt of 916 ppm. Additionally, the selected rock chemistry database also influences the descriptive statistics results. Recently published values for P content within basalt considering GEOROC database are 1309 ppm for median content and 428 ppm for $10^{th}$ percentile and 3186 ppm for $90^{th}$ percentile (Amann and Hartmann, 2019). Thus, before deploying EW to supply nutrients, the chemical composition of rock powder should be known to properly estimate the necessary amount of rock for supplying the demanded nutrients of a specific plant. This would allow to easily estimate the impacts on soil hydrology by the pedotransfer functions of this study or by specific laboratory experiments.

Besides the potential to be used to rejuvenate soil nutrient pools (Leonardos et al., 1987), silicate rock powder can be used to reduce the risk of nitrate mobilization, being indicated for regions in which special care to water preservation is needed.

However, extra input of sodium (Na) to the system, if the rock is rich in this element, could disturb this amelioration effect (Von Wilpert and Lukes, 2003). Besides decreasing nitrate mobilization, co-application of rock powder with other fertilizers can increase the biomass production of crops (Anda et al., 2013;Leonardos et al., 1987;Theodoro et al., 2013).

An additional challenge of the application of rock products will be the assessment of the fate of weathering products, which might be transported eventually into river systems and alter geochemical baselines as evidenced by past land use changes in some large rivers (Hartmann et al., 2007;Raymond and Hamilton, 2018).

## 5. Conclusions

Our results illustrate the potential of Enhanced Weathering (EW) to act as a nutrient source to nutrient demanding AR and BG. This is an important, yet often overlooked, aspect of EW besides $CO_2$ sequestration. The investigated scenarios show that areas with undersupply of P exist, and a C-stock reduction is expected to occur if P is the only limiting nutrient. Considering N and P deficiency together for a low geogenic P supply and high biomass P demand, the C-stock reduction will be up to 59% of the projected total global C sequestration potential of 224 Gt C from the N-unlimited AR scenario. Potential P deficiencies were here based on the soil P availability and P demand scenarios, indicating that the inclusion of P cycles in AR models is necessary to accurately project the C sequestration of forests. Industrial fertilizers can be used to alleviate the P deficiency but the extra input of ammonium along with it can undermine the carbon budget and acidify the soils. Furthermore, acidic soil conditions generally favors the sorption of orthophosphate onto Fe- and Al-(hydro)oxides surfaces and clay minerals, essentially demobilizing P (Shen et al., 2011).

Besides the high chemical P content and relative fast weathering rates, the equilibrated supply of Ca, K, and Mg put the use of basalt powder one step ahead as a potential alternative to industrial fertilizers than other rocks. Regrowth of forests on abandoned agricultural land is a passive landscape restoration method (Bowen et al., 2007). In most of the cases soils become acidic in abandoned agricultural land in the long term (Hesterberg, 1993), which favors the leaching of nutrients (Haynes and Swift, 1986) and heavy metals (Hesterberg, 1993). As a consequence, the regrowth rate of forests might be limited in acidic soils. The use of basalt powder will keep a positive carbon budget, increase the soil pH (Anda et al., 2015;Anda et al., 2009), as basalt powder would act as a buffer maintaining soil pH under neutral to slight alkaline conditions, close nutritional needs of AR and BG, and rock powder can be used to reduce the risk of nitrate mobilization (Von Wilpert and Lukes, 2003). However, to be able to assess the global potential of the combination of land-based biomass NETs with EW, it is necessary to explore related physico-chemical changes of soil influenced by varying EW deployment rates, based on already available data, and then develop improved EW models. They should be tested with field-based approaches. For example, tracking added elements through the ecosystem's soil and plant reservoirs probably needs test sites using advanced methods of nutrient balance and isotope studies, as recently developed (Uhlig et al., 2017;Uhlig and von Blanckenburg, 2019).

In addition to the use for replenishing soil nutrient content, our research suggests that deployment of rock powder on the top soil can enhance Plant Available Water (PAW) for different upper limits. Apart from controlling the nutrient release rates, the texture of deployed rock powder would influence the impacts on soil hydrology together with the initial soil texture. In general, EW appears to have considerable potential for water retention management of top soils. This is an important characteristic not explored before, since under a future scenario of climate change EW can potentially mitigate or alleviate drought effects to a certain extent within areas used for AR and BG plantation. Field and laboratory experiments are needed to quantify soil hydraulic changes under a natural and controlled environment. Besides that, investigation of potential changes of coupling EW with other terrestrial NETs such as Biochar is necessary, since Biochar and EW can increase the amount of soil organic matter, a variable also responsible for increasing water retention of soils.

We show that EW can be an important part of the solution to the problem of nutrient limitation AR and BG might suffer from. Specifically, the potential for hydrological management of soils was shown and it could be used in areas where seasonality and droughts might affect the biomass growth. The use of Enhanced Weathering for hydrological management coupled to land based NETs is worth to investigate. A global management of the carbon pools will need a full ecosystem understanding, addressing nutrient fluxes, and related soil mineralogy changes, soil hydrology, impacts on soil microorganisms, and responses of plants on the diverse array of soil types and climates. Applied ecosystem engineering is likely a future nexus discipline which needs to link local ecosystem processes with a global perspective on carbon pools within a universal effort to manage the carbon cycle.

## 6. ORCID iDs

Wagner de Oliveira Garcia https://orcid.org/0000-0001-9559-0629

Jens Hartmann https://orcid.org/0000-0003-1878-9321

Thorben Amann https://orcid.org/0000-0001-9347-0615

Pete Smith https://orcid.org/0000-0002-3784-1124

Lena R. Boysen http://orcid.org/0000-0002-6671-4984

Daniel Goll https://orcid.org/0000-0001-9246-9671

## 7. Acknowledgements

This study was funded by the German Research Foundation's priority program DFG SPP 1689 on "Climate Engineering–Risks, Challenges and Opportunities?" and specifically the CEMICS2 project. In addition this work was supported by the Deutsche Forschungsgemeinschaft (DFG, German Research Foundation) under Germany´s Excellence Strategy – EXC 2037 'Climate, Climatic Change, and Society' – Project Number: 390683824, contribution to the Center for Earth System Research

and Sustainability (CEN) of Universität Hamburg. DSG is funded by the "IMBALANCE-P" project of the European Research Council (ERC-2013-SyG-610028). We are grateful for the constructive comments and suggestions from the reviewers and editor.

## 8. Review statement.

This paper was edited by Alexey Eliseev and reviewed by Daniel Ibarra and an anonymous referee.

## 9. Author contribution

This article was conceived by the joint work of all authors, which participated in the discussions and writing, with the lead of WOG. The study was designed by W.O.G., J.H., and T.A. W.O.G. compiled all the used data and conducted the calculations. K.K and A.P. supplied the MAgPIE model simulations and the stoichiometric ratios used for bioenergy grass. L.R.B. contributed to handling the JSBACH model outputs and D.G. contributed to the methodology to obtain the P-stock based demand for AR.

## 10. Competing interests

The authors declare that they have no conflict of interest.

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

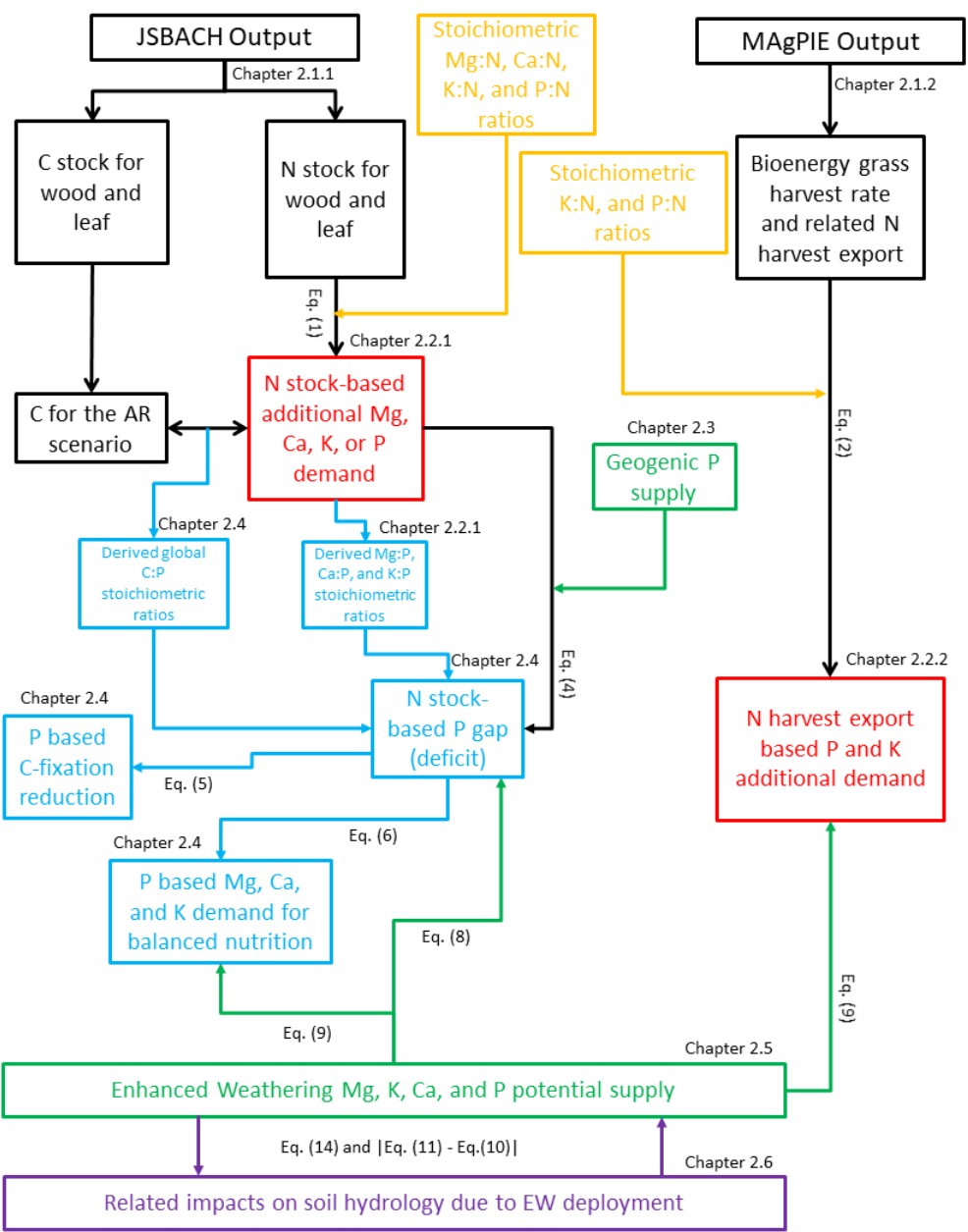


**Fig. 1: Schematic steps and datasets used to derive geogenic nutrient demand from simulated biomass changes, P gaps, reduced C sequestration, and Ca, K, and Mg supply for balanced tree nutrition. Black colors: Outputs from land surface model JSBACH and agricultural production model MAgPIE. Yellow colors: stoichiometric Mg:N, Ca:N, K:N, and P:N ratios used to obtain the N stock-based nutrient demand. Red colors: N stock-based P, Mg, Ca, and K demand for wood and leaf (AR) or N harvest export based P and K demand (BG). Green colors: nutrient supply from geogenic sources (atmospheric P deposition and different soil P pools) or from Enhanced Weathering. Blue colors: derived P gap for AR, derived stoichiometric C:P, Mg:P, Ca:P, and K:P ratios, P based C-fixation reduction, and P based Mg, Ca, and K supply for balanced tree nutrition. Purple colors: Related EW deployment impacts on soil hydrology estimated by pedotransfer functions. AR: Afforestation/reforestation, BG: Bio-energy grass.**

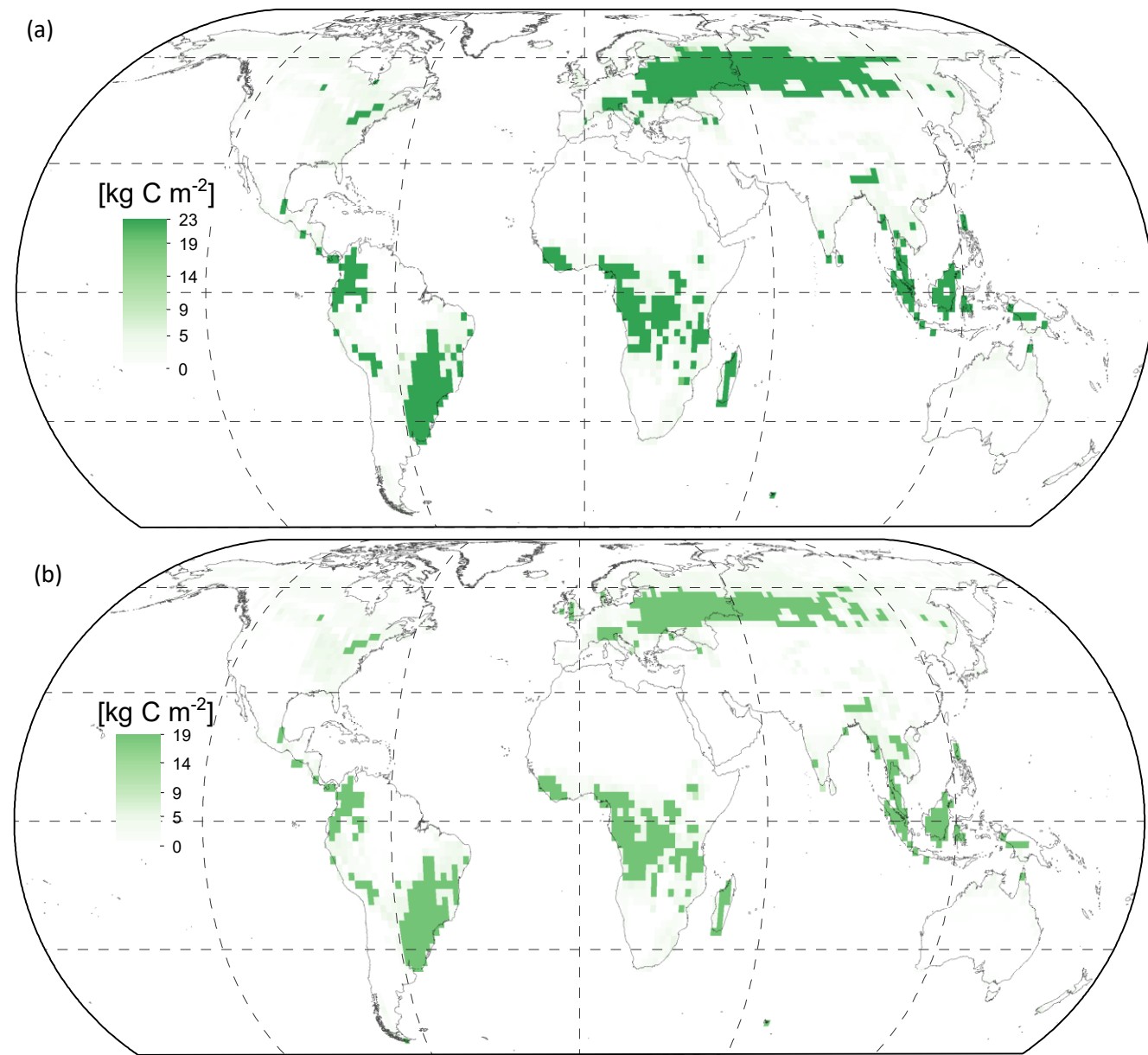


**Fig. 2: Carbon sequestered in different afforestation reforestation scenarios for the 21st century (2006 – 2099) period for a RCP4.5 scenario, according to Kracher (2017). a) For an N-unlimited AR scenario the global C sequestration is 224 Gt C. b) For an N-limited AR scenario the global C sequestration is 190 Gt C. Map generated with ESRI ArcGIS ver. 10.6 (http://www.esri.com).**


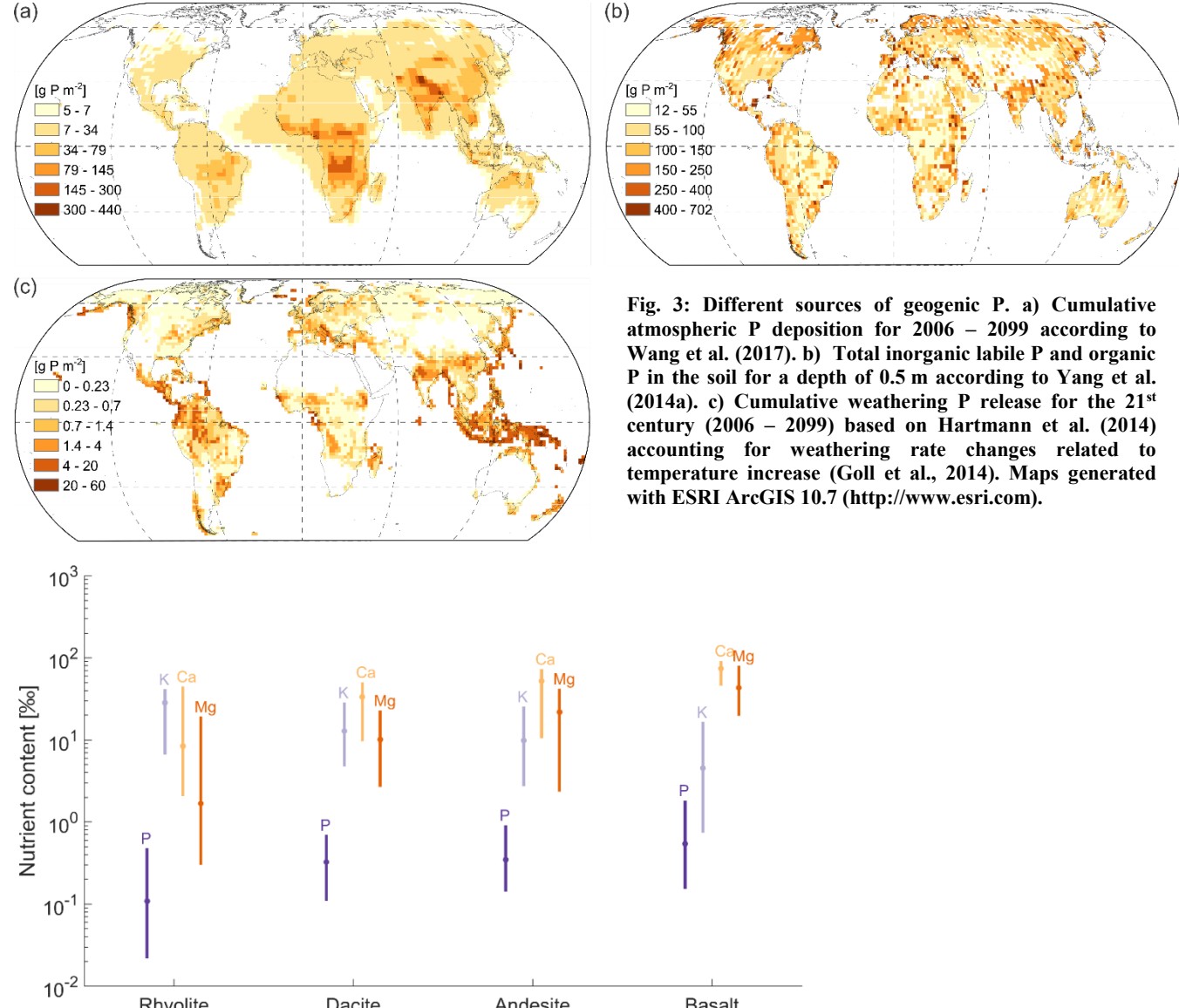

Fig. 3: Different sources of geogenic P. a) Cumulative atmospheric P deposition for 2006 – 2099 according to Wang et al. (2017). b) Total inorganic labile P and organic P in the soil for a depth of 0.5 m according to Yang et al. (2014a). c) Cumulative weathering P release for the 21st century (2006 – 2099) based on Hartmann et al. (2014) accounting for weathering rate changes related to temperature increase (Goll et al., 2014). Maps generated with ESRI ArcGIS 10.7 (http://www.esri.com).

Fig. 4: Statistical data of major element concentration in rocks, median values (filled circles) and range (5th and 95th percentiles, whiskers). Values from Earthchem webportal (http://www.earthchem.org). The number of chemical analysis used to calculate the descriptive statistics were: 2985 chemical analysis for rhyolite, 3008 chemical analysis for dacite, 11099 chemical analysis for andesite, and 23816 chemical analysis for basalt.


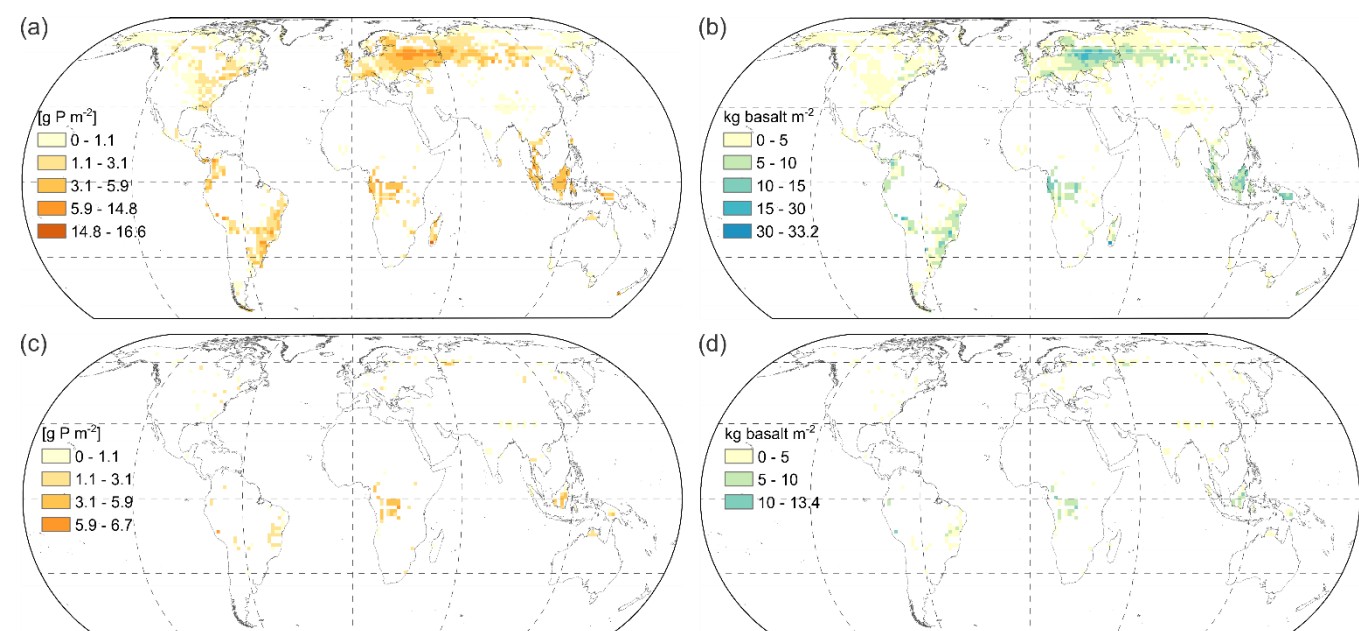

Fig. 5: Areas with potential P gap for the nutrient budget of the N-limited AR scenario (after 94 years of simulation) assuming P concentrations within foliar and wood material corresponding to mean values (Table 1). a) Geogenic P supply scenario one (geogenic P from weathering plus atmospheric P deposition as source of P). b) Basalt deployment necessary to close P gaps from P budget scenario of Fig. 5a. c) Geogenic P supply scenario two (geogenic P from soil inorganic labile P and organic P pools plus atmospheric P deposition and P from weathering as source of P). d) Basalt deployment necessary to close P gaps from P budget scenario of Fig. 5c. Map generated with ESRI ArcGIS 10.7 (http://www.esri.com).

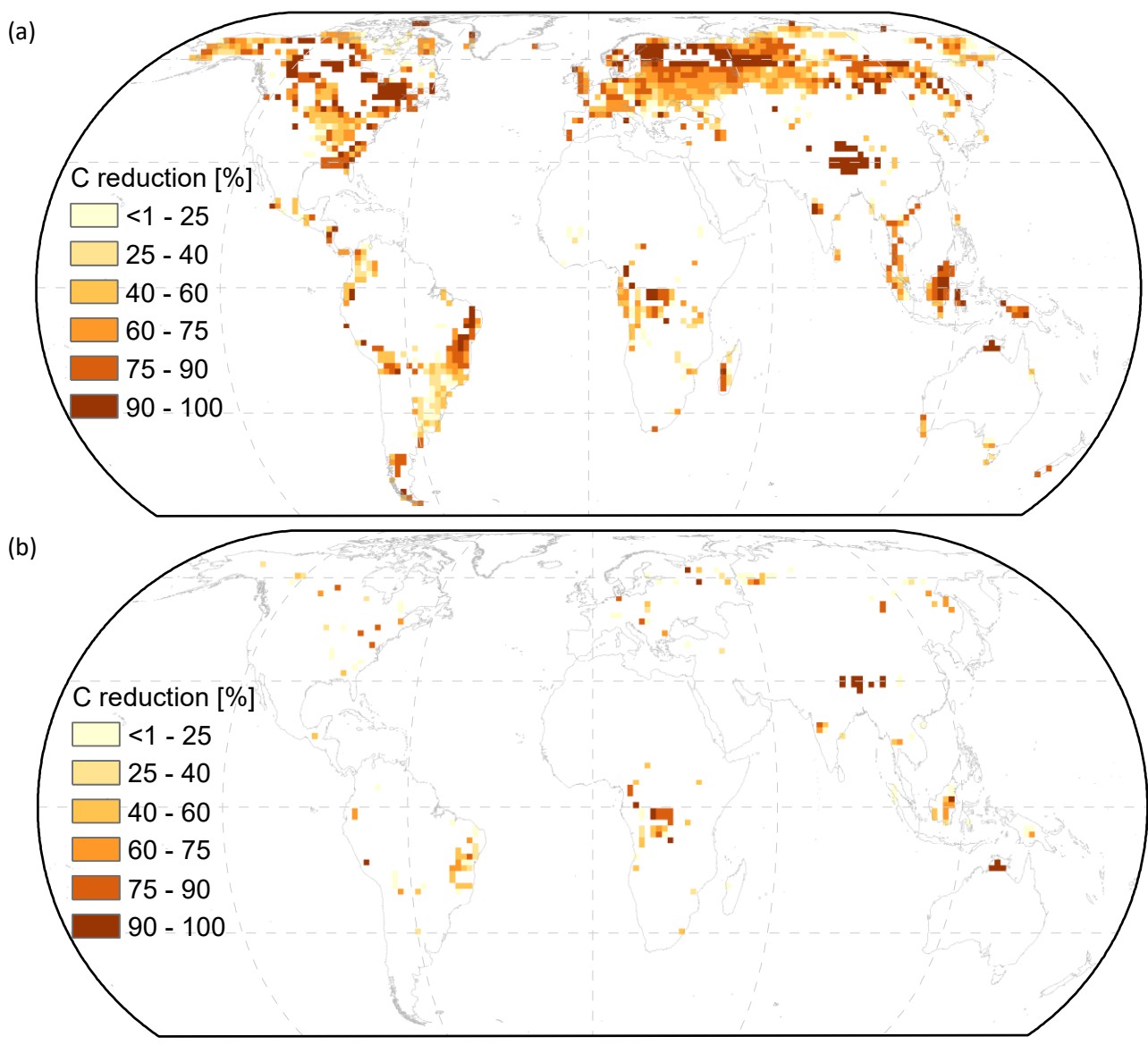

(a)

C reduction [%]
- <1 - 25
- 25 - 40
- 40 - 60
- 60 - 75
- 75 - 90
- 90 - 100

(b)

C reduction [%]
- <1 - 25
- 25 - 40
- 40 - 60
- 60 - 75
- 75 - 90
- 90 - 100

**Fig. 6: Reduction on forest C sequestration due to geogenic P limitation. C-reduction estimated from stoichiometric C:P ratios for the N-limited AR scenario assuming P concentrations within foliar and wood material corresponding to mean values (Table 1). On Fig. 2b we present the C sequestration potential if geogenic P supply is not limiting biomass growth. a) C-reduction based on P gaps of Fig. 5a, obtained for geogenic P supply scenario one (geogenic P from weathering plus atmospheric P deposition as source of P). b) C-reduction based on P gaps of Fig. 5c, obtained for geogenic P supply scenario two (geogenic P from soil inorganic labile P and organic P pools plus atmospheric P deposition and P from weathering as source of P). For resulting global C reduction check Table 3. Map generated with ESRI ArcGIS 10.7 (http://www.esri.com).**



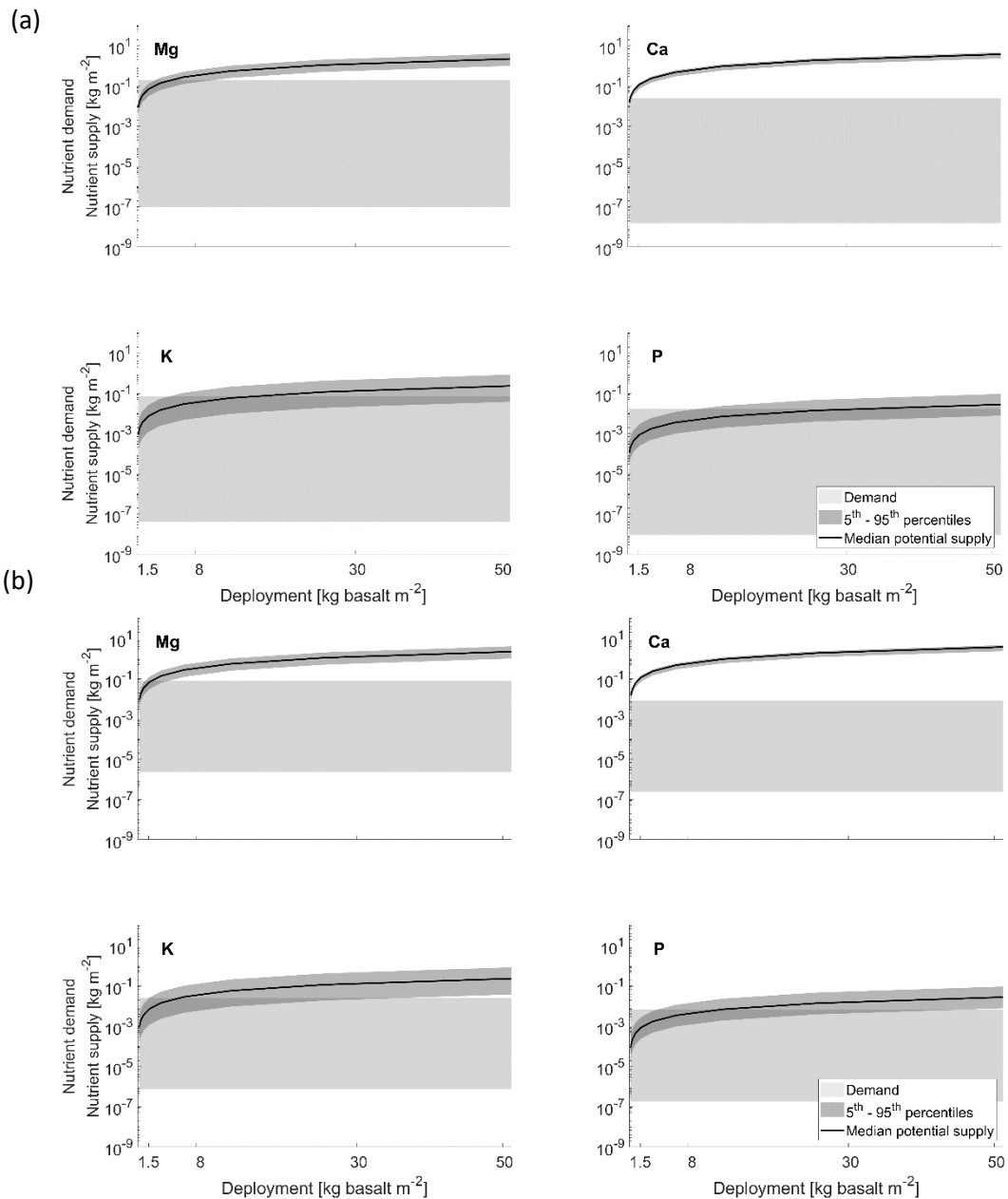

**Fig. 7: Mg, Ca, K, and P supply by basalt dissolution (logarithmic curve) given as median and ranges (5th and 95th percentiles) (dark grey areas). Horizontal filled boxes indicate the nutrient demand for maximum (17.1 g P m-2) and minimum (<<1 g P m-2) gap of each geogenic P supply scenario for P and derived Mg, Ca, and K demand for balanced tree nutrition assuming mean foliar and wood material chemistry (Table 1). a) Based on minimum and maximum P gap values of <1 g P m-2 and 16.6 g P m-2, which were obtained for a geogenic P supply scenario one (geogenic P from weathering plus atmospheric P deposition as source of P). b) Based on minimum and maximum P gap values of <1 g P m-2 and 6.7 g P m-2, which were obtained for a geogenic P supply scenario two (geogenic P from soil inorganic labile P and organic P pools plus atmospheric P deposition and P from weathering as source of P).**

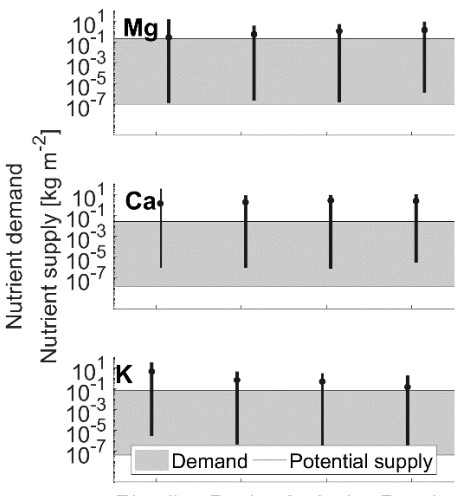

**Fig. 8: Potential macronutrient (Mg, Ca, and K) supply of different rocks for closing projected P gaps of <<1 to 17.1 g P m$^{-2}$. Median and ranges (5$^{th}$ and 95$^{th}$ percentiles) of potential supply based on rock chemistry.**

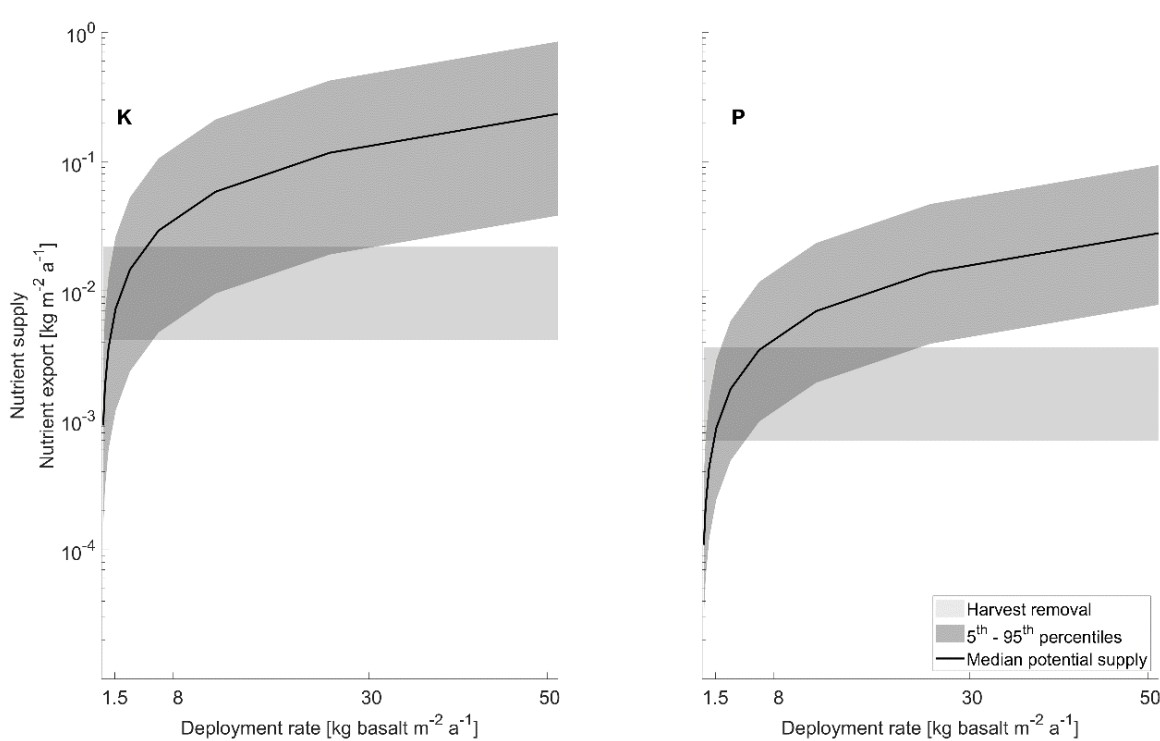

**Fig. 9: Projected K and P supply (logarithmic curve) by basalt dissolution given as median ranges (5$^{th}$ and 95$^{th}$ percentiles) for bio-energy grasses K and P demand (horizontal filled boxes) based on global minimum 0.7 kg m$^{-2}$ a$^{-1}$ and maximum 3.6 kg m$^{-2}$ a$^{-1}$ harvest rates for simulation years of 1995 – 2090. The amount of exported nutrients by several harvest rates higher than the minimum and lower than the maximum harvest rates are represented by the vertical filled boxes.**

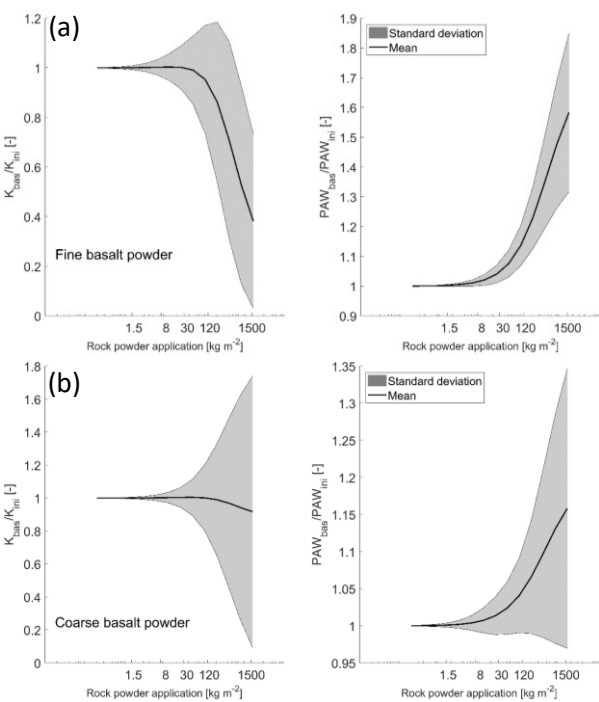

Fig. 10: Relative impacts on soil saturated hydraulic conductivity ($K_S$) and Plant Available Water (PAW). $K_{bas}$ and $PAW_{bas}$ respectively represent the estimated soil $K_S$ and PAW after basalt application. $K_{ini}$ is the estimated initial soil $K_S$ and $PAW_{ini}$ is the estimated initial PAW of different soils. a) Application of a fine basalt texture (15.6% clay, 83.8% silt, and 0.6% fine sand). b) Application of a coarse basalt texture (15.6% clay, 53.8% silt, and 30.6% fine sand) for areas corresponding to P gaps of geogenic P supply scenario one, for the N-unlimited AR scenario (Supplement S1 Fig. S7a). Mean and standard deviations for n=15318 grid cells. cf., Supplement S1 section D for impacts on initial $K_S$ and PAW of fine or coarse basalt powder texture on soils of P gap areas from (Supplement S1 Fig. S7c).


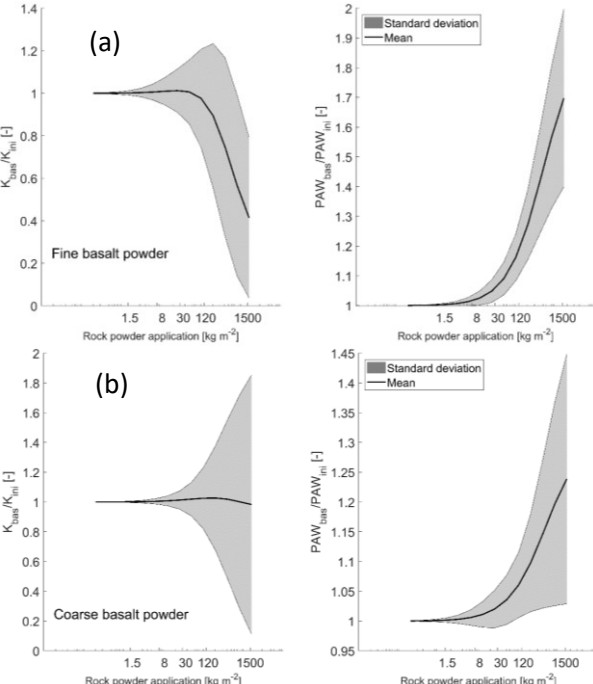

Fig. 11: Relative impacts on soil saturated hydraulic conductivity ($K_S$) and Plant Available Water (PAW). $K_{bas}$ and $PAW_{bas}$ respectively represents the estimated soil $K_S$ and PAW after basalt application. $K_{ini}$ is the estimated initial soil $K_S$ and $PAW_{ini}$ is the estimated initial PAW of different soils. a) Application of a fine basalt texture (15.6% clay, 83.8% silt, and 0.6% fine sand). b) Application of a coarse basalt texture (15.6% clay, 53.8% silt, and 30.6% fine sand) for areas corresponding to P budget scenario two, for the N-unlimited AR scenario (Supplement S1 Fig. S7c). Mean and standard deviations for n=2525 grid cells.


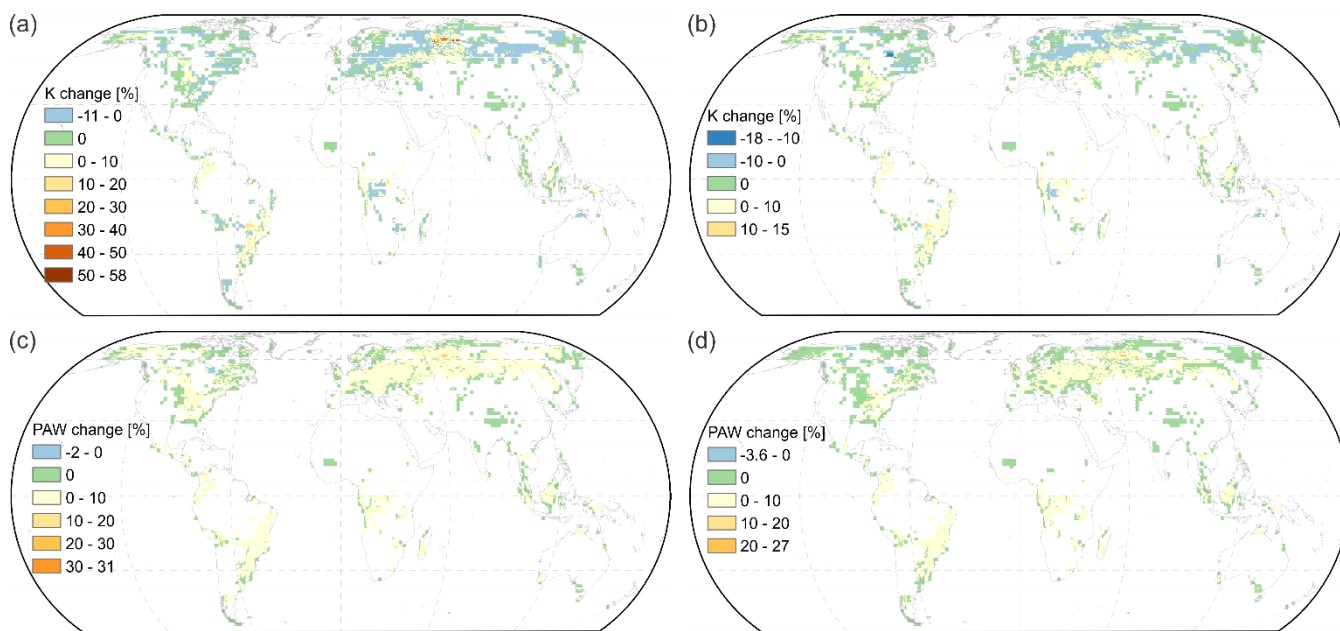

**Fig. 12: Impacts on soil hydrology estimated according to Saxton and Rawls (2006) equations for basalt deployment mass coincident to areas with potential P gap for the nutrient budget of the N-unlimited AR scenario assuming P concentrations within foliar and wood material corresponding to mean values (Supplement Fig. S7a). a) Hydraulic conductivity (K) changes relative to initial soil values for a fine basalt texture (15.6% clay, 83.8% silt, and 0.6% sand) being deployed. b) Hydraulic conductivity (K) changes relative to initial soil values for a coarse basalt texture (15.6% clay, 53.8% silt, and 30.6% fine sand) being deployed. c) Plant available water (PAW) changes relative to initial soil values for a fine basalt texture (15.6% clay, 83.8% silt, and 0.6% sand) being deployed. d) Plant available water (PAW) changes relative to initial soil values for a coarse basalt texture (15.6% clay, 53.8% silt, and 30.6% fine sand) being deployed. Map generated with ESRI ArcGIS 10.7 (http://www.esri.com).**



**Table 1: Stoichiometric parameters for different pools and biomes used in this study.**

| Biome | | Tropical evergreen | | | | | Tropical deciduous | | | | | Temperate evergreen | | | | |
|---|---|---|---|---|---|---|---|---|---|---|---|---|---|---|---|---|
| Leaf[a] | | mean | Std | n | 5th percentile | 95th percentile | mean | Std | n | 5th percentile | 95th percentile | mean | Std | n | 5th percentile | 95th percentile |
| C:N | | 29.72 | 15.01 | 4 | 16.33 | 46.49 | 26.96* | 10.53* | 171* | 14.50* | 46.7* | 49.11 | 12.15 | 8 | 33.54 | 65.69 |
| P:N | | 0.06 | 0.02 | 59 | 0.04 | 0.10 | 0.07 | 0.03 | 43 | 0.04 | 0.13 | 0.09 | 0.03 | 23 | 0.05 | 0.13 |
| K:N | [-] | 0.97 | 0.80 | 2 | 0.46 | 1.48 | 1.26 | 0.93 | 22 | 0.23 | 2.45 | 0.47 | 0.09 | 12 | 0.33 | 0.58 |
| Ca:N | | 2.73 | 3.44 | 2 | 0.54 | 4.91 | 1.55 | 0.78 | 22 | 0.52 | 2.90 | 0.73* | 0.67* | 150* | 0.16* | 1.94* |
| Mg:N | | 0.40 | 0.52 | 2 | 0.07 | 0.73 | 0.37 | 0.29 | 22 | 0.10 | 0.83 | 0.21* | 0.21* | 115* | 0.05* | 0.66* |
| Wood[b] | | mean | Std | n | 5th percentile | 95th percentile | mean | Std | n | 5th percentile | 95th percentile | mean | Std | n | 5th percentile | 95th percentile |
| C:N | | 235 | 244 | 9 | 56 | 610 | 235 | 244 | 9 | 56 | 610 | 235 | 244 | 9 | 56 | 610 |
| P:N | | 0.15 | 0.20 | 684 | 0.04 | 0.30 | 0.15 | 0.20 | 684 | 0.04 | 0.30 | 0.15 | 0.20 | 684 | 0.04 | 0.30 |
| K:N | [-] | 0.60 | 0.40 | 700 | 0.20 | 1.20 | 0.60 | 0.40 | 700 | 0.20 | 1.20 | 0.60 | 0.40 | 700 | 0.20 | 1.20 |
| Ca:N | | 1.80 | 1.30 | 705 | 0.40 | 4.30 | 1.80 | 1.30 | 705 | 0.40 | 4.30 | 1.80 | 1.30 | 705 | 0.40 | 4.30 |
| Mg:N | | 0.20 | 0.10 | 681 | 0.10 | 0.40 | 0.20 | 0.10 | 681 | 0.10 | 0.40 | 0.20 | 0.10 | 681 | 0.10 | 0.40 |

| Biome | | Temperate deciduous | | | | | Shrubs raingreen | | | | | Shrubs deciduous | | | | |
|---|---|---|---|---|---|---|---|---|---|---|---|---|---|---|---|---|
| Leaf[a] | | mean | Std | n | 5th percentile | 95th percentile | mean | Std | n | 5th percentile | 95th percentile | mean | Std | n | 5th percentile | 95th percentile |
| C:N | | 55.30 | 12.02 | 2 | 47.65 | 62.95 | 26.31 | 6.83 | 2 | 21.97 | 30.65 | 26.96* | 10.53* | 171* | 14.5* | 46.70* |
| P:N | | 0.08 | 0.03 | 32 | 0.04 | 0.13 | 0.07 | 0.01 | 2 | 0.06 | 0.08 | 0.08* | 0.05* | 662* | 0.04* | 0.16* |
| K:N | [-] | 0.43 | 0.13 | 23 | 0.24 | 0.61 | 0.38 | 0.02 | 2 | 0.37 | 0.39 | 0.59* | 0.45* | 207* | 0.24* | 1.50* |
| Ca:N | | 0.73* | 0.67* | 150* | 0.16* | 1.94* | 0.44 | 0.08 | 2 | 0.39 | 0.50 | 0.73* | 0.67* | 150* | 0.16* | 1.94* |
| Mg:N | | 0.21* | 0.21* | 115* | 0.05* | 0.66* | 0.09 | 0.04 | 2 | 0.06 | 0.12 | 0.21* | 0.21* | 115* | 0.05* | 0.66* |
| Wood[b] | | mean | Std | n | 5th percentile | 95th percentile | mean | Std | n | 5th percentile | 95th percentile | mean | Std | n | 5th percentile | 95th percentile |
| C:N | | 235 | 244 | 9 | 56 | 610 | 235 | 244 | 9 | 56 | 610 | 235 | 244 | 9 | 56 | 610 |
| P:N | | 0.15 | 0.20 | 684 | 0.04 | 0.30 | 0.15 | 0.20 | 684 | 0.04 | 0.30 | 0.15 | 0.20 | 684 | 0.04 | 0.30 |
| K:N | [-] | 0.60 | 0.40 | 700 | 0.20 | 1.20 | 0.60 | 0.40 | 700 | 0.20 | 1.20 | 0.60 | 0.40 | 700 | 0.20 | 1.20 |
| Ca:N | | 1.80 | 1.30 | 705 | 0.40 | 4.30 | 1.80 | 1.30 | 705 | 0.40 | 4.30 | 1.80 | 1.30 | 705 | 0.40 | 4.30 |
| Mg:N | | 0.20 | 0.10 | 681 | 0.10 | 0.40 | 0.20 | 0.10 | 681 | 0.10 | 0.40 | 0.20 | 0.10 | 681 | 0.10 | 0.40 |

*Values obtained from all biomes. [a] Stoichiometric ratios derived from a global leaf chemistry database (Vergutz et al., 2012). [b] Stoichiometric ratios derived from a US soft- and hardwood database (Pardo et al., 2005). cf., SI-table.xlsx file for used database.

**Table 2: Geogenic P sources used for each geogenic P supply scenario.**

| P source | Resolution | Geogenic P supply scenario one | Geogenic P supply scenario two | Reference |
|---|---|---|---|---|
| Soil organic P and inorganic labile P | 0.5° | | X | (Yang et al., 2014a) |
| Atmospheric P deposition | 1° | X | X | (Wang et al., 2017) |
| P from weathering | 1 km$^2$ | X | X | (Hartmann et al., 2014) |


**Table 3: Global P gap, maximum estimated P gap, maximum C sequestration reduction, and global C reduction for the natural N supply (N-limited) AR scenario (projected C sequestration of 190 Gt C).**

| N supply | Geogenic P supply | Maximum estimated P gap [g P m$^{-2}$] | | | Global P gap [Mt P] | | | Maximum C sequestration reduction [kg C m$^{-2}$] | | | Global C reduction [Gt C] | | |
|---|---|---|---|---|---|---|---|---|---|---|---|---|---|
| | | Wood and leaves P content | | | | | | | | | | | |
| | | 5th percentile | Mean | 95th percentile | 5th percentile | Mean | 95th percentile | 5th percentile | mean | 95th percentile | 5th percentile | mean | 95th percentile |
| Limited | Scenario one | 4.1 | 16.6 | 30.2 | 9.2 | 76.6 | 181.0 | 9.7 | 14.5 | 15.6 | 23.0 | 71.0 | 98.0 |
| | Scenario two | 1.6 | 6.7 | 12.2 | 1.0 | 9.9 | 34.7 | 4.7 | 6.2 | 6.5 | 3.0 | 9.5 | 19.0 |

**Table 4: Minimum and maximum soil hydraulic conductivity for areas coincident to the P gap areas of each geogenic P supply scenario one, for the N-unlimited AR scenario (Supplement S1 Fig. S7a).**

| | Geogenic P supply scenario one | Geogenic P supply scenario two |
|---|---|---|
| Hydraulic conductivity (K) [m s$^{-1}$] | | |
| Min | $1.5 \times 10^{-7}$ | $2.7 \times 10^{-7}$ |
| Max | $1.7 \times 10^{-4}$ | $7.8 \times 10^{-5}$ |
| Plant available water (PAW) [%] | | |
| Min | 4 | 6 |
| Max | 32 | 28 |