# Peer review of "Impacts of Enhanced Weathering on biomass production for negative emission technologies and soil hydrology"

_Biogeosciences, 2019_

## Referee Comment (RC1) · Daniel Ibarra (Referee) · 8 Jan 2020

This study assesses the impact of enhanced weathering on biomass production as applied for negative emissions and looks at the global scale impact on soil hydrology. The calculations and modeling conducted is thorough, though it was somewhat difficult to follow all of the calculations, which required flipping to the supplement. The study's findings/calculations have large (honest) error envelops but they show that enhanced weathering could close the projected P gaps of the AR scenario they analyze, decreasing or replacing the use of fertilizers. Further, the authors contend that enhanced weathering has positive impacts on nutrients, soil pH and soil (micro/macro) nutrient

pools, while also impacting the plant available water capacity.

Overall I found the manuscript well written (comments below) but I have several structural comments: 1) The combination of "results and discussion" in the second half of the manuscript is very difficult to follow. In particular, I thought this section was pretty much a "discussion and implications" section rather than demonstrating the key findings of this work. I'd ask the authors to consider writing a new concise and clear "results" section clearly outlining what was found with the modeling and calculations, stepping through the calculations, describing the maps in Figures 2 and 3, and the curves int he subsequent figures, as well as the tables. 2) Some important details are buried in the supplement. In particular, I think that portions of S3-S5 sections should be included in more detail in the main text. The manuscript is currently not very long and I think that this would greatly enhance the explanation of what was done in the main text.

I do not think these reorganization suggestions or changes suggested below are very substantial and thus recommend minor revision.

Comments 62: How does this older Gaillardet number compare to those more recently produced by co-author Hartmann's papers and Moon et al. (2014)? I would consider citing all of these sources and giving a range of the $CO_2$ consumption rate by silicate weathering is.

71-76: Work by Porder and Hilley, as well as Waldbauer and Chamberlain (2005) should be cited in this paragraph or previously.

97: Have others done this type of analysis or suggested this since von Liebig and Playfair (1843)?

108: What about changes in precipitation and runoff? Relatedly, what are the "Parameters" in yellow in Figure 1? It might help to list them on the figure.

169: In addition to Figure 1, a schematic of what is actually being done in the methods

would be very helpful. I'm not sure the flow chart fully captures the processes being described in this section.

191: Again, what about changes in runoff to weathering rate? What is this 9%/degC equivalent to in an Arrhenius style calculation? While temperature increases make sense (kinetics) much of the world weathers chemostatically (equilibirium, see Winnick and Maher (2018, EPSL) and papers by Godsey for example) so changes in soil infiltration rate (P-ET = runoff) and thus soil moisture should also influence these changes.

220-233: What about the grain size of the rock powder be applied? Does that matter at all? I assume it would be based on reactive transport simulations such as those by Maher (2010, 2011) for example. And relatedly how fine the rock is powdered will change the texture described in the next section right?

240-243: Thus, this scenario will be the maximum effect? I think that should be stated explicitly here and elsewhere.

248 (equation 10) and text surrounding: Is this the 'pedotransfer function' mentioned in the above paragraph? Is the only empirical values the lambda? Further, where does the 1930 in the equation come from? A lot of this appears to be buried in the supplement, maybe it'd be more helpful if it is more explicitly stated that this is from Saxton and Rawls (2006) upfront in this section.

332: What about other soil types? Is this information in the literature? What is the most common agricultural soil type?

379: But if they are empirical wouldn't those datasets inherently have clay minerals forming in those systems? In general, I think the caveats brought up in this paragraph should be acknowledged earlier in the paper.

Specific Line by Line Comments 19,21 and 22: Are these ranges 95%/2 sigma ranges? Lines 147 and 172 suggests that this might be driven by the minimum and maximum harvest rates. What is the central estimate? Abstract should be able to stand on it's

own so please specify in parentheticals and provide the median or mean estimate.

33 and 34: Please put "i.e." and "e.g." phrases in parentheticals

59: Suggest authors but "i.e., P, Mg, Ca and K" in the parenthetical before the citation. I also think some more recent work should be cited here and in the next line (60) for controls on atmospheric $CO_2$ over geologic timescales.

185: Are the databases sources at most 2 by 2 degrees?

196: Suggest putting "cf. Wang et al. (2017)..." as a parenthetical

223: Please add citation and average P content in basalt here.

262: "here estimated" is confusing.

282-283: Is this a citation in the parenthetical?

307: "under study"?

387: please fix grammar of this sentence

442: I would suggest citing Uhlig's papers here rather than what I believe from the reference list list is the PhD thesis. The papers are Uhlig et al. (2017, Biogeosciences) and Uhlig and von Blanckenburg (2019, Frontiers in Earth Science).

---

## Referee Comment (RC2) · Anonymous Referee #2 · 23 Jan 2020

General comments

This study explores the potential of enhanced weathering to fill the gap in phosphorus needed in order to sustain negative emission technologies such as afforestation/reforestation and energy production from bioenergy grass. It shows that the deployment of enhanced weathering, in terms of basalt addition to the ground, will have beneficial effects on biomass production by supplying phosphorus and other nutrients, in addition to the direct effect of enhancing the $CO_2$ consumption by silicate weathering. The paper is mostly well written and fits the scope of Biogeosciences. I have a number of comments that should be addressed before the paper is suitable for publication.

Major comments

A number of uncertainties are considered in the paper, but it seems that a lot of weight is given to some uncertainties, while other potentially large uncertainties are neglected. For example, only a single $CO_2$/climate scenario is considered and for this particular scenario only results from a single afforestation/bioenergy model is used. What was the reason to choose this particular climate scenario? A discussion on the applicability of the conclusions drawn from the paper to alternative scenarios should be included. For the carbon sequestration from afforestation you mention (line 260) that the available estimates of carbon uptake vary between 0.3 and 3.3 GtC/a, while in the paper a value of 2 GtC/a is used. On the other hand, a lot of weight in the uncertainty is given to e.g. the P concentration in basalt (5-95th percentiles). To my understanding it seems that this uncertainty is not so relevant for the present study, as I assume that for the use in EW basalt with relatively high P concentrations could in principle be selected. Would the interquartile range possible provide a more appropriate measure of the uncertainty in this parameter? In general in the paper it is difficult to immediately associate the uncertainty ranges to uncertainty sources.

For AR it seems that the largest uncertainty is related to the geogenic P-supply, with scenario two showing basically no P limitation. Is there really no observation-based evidence to suggest that one or the other scenario is more realistic? This is a fundamental question for the purpose of the paper, because if P limitation is not an issue the benefit of EW in these areas will be limited to the direct $CO_2$ consumption by weathering.

In the main text there are many references to supplementary materials. References to supplementary material should be limited as much as possible for a better readability of the paper, considering also that papers in Biogeosciences are not subject to strict length limitations. Since the main paper contains only relatively few figures, I would suggest to move some of the figures from the supplementary to the main paper. For

example Figs S1,S2,S3 could be merged into one figure and added to the paper. It would help to get an idea of the numbers that could then be more easily compared with e.g. the P gap in Fig. 2. I would also strongly suggest to add Fig. S8 to the main paper.

The results for the N-unlimited scenario, which are not discussed in the main text, could be presented in a separate section in the supplementary material. It is very confusing to find figures from N-unlimited experiments mixed inbetween N-limited figures.

Minor comments

Figures should be numbered progressively according to where they are referenced in the text (i.e. Fig. 1 should be referenced before Fig. 2). This is valid also for figures in the supplementary.

Line 21-22: you need to mention what scenario you are considering here, otherwise the numbers make no sense since they will be strongly dependent on future CO2/climate evolution.

Line 23: 'K' not defined here

Line 22: it would be helpful if the same unit would be used, either GtC or GtCO2. Otherwise it is difficult to compare the numbers.

Line 95: '...we discuss only the N-limited AR scenario...'

Line 119: please check this sentence

Line 147: are these numbers global averages for the areas with bioenergy plantations?

Line 186: nearest neighbour interpolation does not seem to be a very good interpolation choice for very high resolution input data (e.g. geogenic P release rates). A single 1 km$^2$ value will not be very representative for a 2x2$°$ cell.

Line 425: where does this number come from?

---

## Author Comment (AC2) · 20 Feb 2020

Dear reviewer, we have carefully reviewed the comments and revised the manuscript. We thank you for constructive remarks and took care to include all of them in the manuscript. We added a detailed reply to each point as addressed below.

**Reviewers comment**

Our reply

**RC1: Comments 62: How does this older Gaillardet number compare to those more recently produced by co-author Hartmann's papers and Moon et al. (2014)?**

**I would consider citing all of these sources and giving a range of the CO2 consumption rate by silicate weathering is**

In line 67 and 68 we gave the range and considered the suggested authors.

**RC1: 71-76: Work by Porder and Hilley, as well as Waldbauer and Chamberlain (2005) should be cited in this paragraph or previously**

The work of Porder and Hilley (2011) is cited in line 80, and Waldbauer and Chamberlain (2005) is cited in line 64.

**RC1: 97: Have others done this type of analysis or suggested this since von Liebig and Playfair (1843)?**

We rephrased the sentence. According to Liebig's Law of the minimum(von Liebig and Playfair, 1843), biomass productivity will be limited by the scarcest nutrient in the soil. This law is widely applied to biological populations and in many ecosystem models. The new sentence is in line 103 to 106.

**RC1: 108: What about changes in precipitation and runoff?**

The weathering rates of minerals are controlled by the following mechanisms: (i) soil pH (Lasaga et al., 1994;Grathwohl, 2014;Arshad et al., 1972); (ii) redox conditions for Fe and Mn minerals (Hering and Stum, 1990;Gilkes et al., 1973;Cánovas et al., 2019); (iii) soil solution composition (Lasaga et al., 1994;Grathwohl, 2014;Hayes et al., 2020) and soil moisture (Sverdrup et al., 1995;Appelo and Postma, 2005); (iv) temperature (Lasaga et al., 1994;Velbel et al., 1990;Hayes et al., 2020); and (v) grain size/density of exposed structural defects on mineral surfaces (Lasaga et al., 1994;Arshad et al., 1972;Holdren Jr and Speyer, 1985;Cánovas et al., 2019). In this study we did not account for impact of changes in hydrology on weathering because model projections of hydrology in the future are not reliable. i.e. for most parts of the global model predictions it differs from each other despite similar boundary conditions (e.g. IPCC AR5). Model predictions of temperature change, on the other hand, are more reliable and all

models agree on the sign (i.e. warming). Goll et al. (2014) used a simple weathering model that accounts for temperature, runoff and soil thickness controls on weathering, and illustrated that global warming was the dominating driver behind changes in weathering using climate change reconstructions from four earth system models. The reconstructions showed large differences with respect to hydrology and changes in global weathering rates to changes in temperature were rather minor. The temperature sensitivity of global weathering rates was significant and rather homogenous among four different climate models. The temperature sensitivity implicitly accounts for changes in hydrology, however it is not able to capture regional differences. Thus we only account for the T changes. This was done earlier in Sun et al. (2017) Earth's Future. Besides the large uncertainty with respect to climate change projections, current data availability is insufficient to calibrate a complex global model, which are currently only applied on site to catchment scale (Pierret et al., 2018;Ackerer et al., 2018;Beaulieu et al., 2016). We argue that a simple model of weathering is appropriate. The used model is very simplistic but the used weathering rates were calibrated on 381 catchments in Japan (Hartmann et al., 2009) and describes the chemical weathering as a function of runoff and lithology, temperature and soil thickness (Hartmann et al., 2014). The model is increasingly used for global scale studies (Sun et al., 2017;Goll et al., 2014;Wang et al., 2018). We included part of this discussion in line 216 to line 222.

**RC1: Relatedly, what are the "Parameters" in yellow in Figure 1? It might help to list them on the figure**

We rearranged Figure 1 and they are explicitly specified there.

**RC1: 169: In addition to Figure 1, a schematic of what is actually being done in the methods own so please specify in parentheticals and provide the median or mean estimate**

We rearranged Figure 1, so the used steps are clearer and they refer to the respective chapter sections and used equations.

**RC1: 191: Again, what about changes in runoff to weathering rate?**

See above.

**RC1: What is this 9%/degC equivalent to in an Arrhenius style calculation? While temperature increases make sense (kinetics) much of the world weathers chemostatically (equilibirium, see Winnick and Maher (2018, EPSL) and papers by Godsey for example) so changes in soil infiltration rate (P-ET = runoff) and thus soil moisture should also influence these changes**

The Arrhenius corresponding average increase is 6.5%/degC for basalt that had an average activation energy of 50 kJ/mol, 3%/degC for carbonate activation energy of 24 kJ/mol, and 10%/degC for a felsic rock type activation energy of 74 kJ/mol (Fig. 1). Our 9%/degC value is also based on the Arrhenius calculation but represents the average value considering different lithotypes (they had mean apparent activation energies ranging from 50 to 80 kJ/mol and a deviation of $\pm$ 20 kJ/mol, based on field data and laboratory experiments (Goll et al., 2014)). The apparent temperature sensitivity of weathering from Goll et al. (2014) implicitly includes changes in hydrology as it is derived from global model simulations using a model which describes the chemical weathering as a function of runoff and lithology, temperature and soil thickness (Hartmann et al., 2014). As lay down above we omit a more detailed representation of water effects due to high uncertainty with respect to projections of hydrology (IPCC AR5) and as temperature was found to be the dominant driver of changes (Goll et al., 2014).

**RC1: 220-233: What about the grain size of the rock powder be applied? Does that matter at all? I assume it would be based on reactive transport simulations such as those by Maher (2010, 2011) for example. And relatedly how fine the rock is powdered will change the texture described in the next section right?**

Yes, the grain size matters since it will influence on the exposed reactive surface area (generally fine textured basalt powder would have a high reactive surface area than a coarser textured basalt powder) and consequently the weathering rates. However,

in our approach we considered that the grain sizes would range in between 0.6 – 90 $\mu$m (for the fine basalt powder) which is enough to completely dissolve the deployed rock powder after one year (Strefler et al., 2018). For the coarse basalt powder 70% of its granulometry fall into the 0.6 – 90 $\mu$m range and from the other 30%, at least 20% would be dissolved in one year (Strefler et al., 2018). The finest grain size we can consider is the clay size which comprehends grains >1 $\mu$m and <3.9 $\mu$m. If a basalt powder would have only clay size granulometry, the effects on soil hydraulic conductivity would decrease in average by 37% for deployment amount of 30 kg basalt m-2 (for the same deployment amount and for the fine rock powder used in our work, the hydraulic conductivity would decrease by 2%). However, the finer the grain gets the higher the energy input for grinding is, which can drastically affect the costs of EW (it can reach up to 500USD/tCO2 sequestered; Strefler et al., 2018). Part of this discussion is included in line 254 to 255. Additionally, we make it clear that the used the grain sizes not higher than 90 $\mu$m in line 279 to 283 and discussed about the assumptions of full dissolution in chapter 4.4 in line 562.

**RC1: 240-243: Thus, this scenario will be the maximum effect? I think that should be stated explicitly here and elsewhere**

Thanks, now it is explicitly stated in lines 283, 285, 393, and 394.

**RC1: 248 (equation 10) and text surrounding: Is this the 'pedotransfer function' mentioned in the above paragraph? Is the only empirical values the lambda? Further, where does the 1930 in the equation come from? A lot of this appears to be buried in the supplement, maybe it'd be more helpful if it is more explicitly stated that this is from Saxton and Rawls (2006) upfront in this section**

This is one of the used pedotransfer functions. The 1930 is a constant (i.e., a regression coefficient) which is dependent on the database used to derive the pedotransfer equation as described in the work from (Campbell, 1974). Now the pedotransfer equations are present in the main text in subchapter 2.6. No, lambda is estimated according
to equation 18. The empirical values are the organic matter content and the granulometry of soil and rock powder (equations 15-17).

**RC1: 332: What about other soil types? Is this information in the literature? What is the most common agricultural soil type?**

Considering the extension, Oxisols are the most common agricultural soil type followed by Ultisols. We added the value for Ultisol in line 450 to 453.

**RC1: 379: But if they are empirical wouldn't those datasets inherently have clay minerals forming in those systems? In general, I think the caveats brought up in this paragraph should be acknowledged earlier in the paper**

The used databases probably contained clay minerals in the clay-sized fraction of the soils. But the Saxton and Rawls (2006) equations accounted only for soil texture and soil organic matter content influences on water retention and hydraulic conductivity. The sentence was misleading and we changed it. You can find the improved sentence in line 523 to 531. What was meant by the old sentence was that once the rock powder will be added into the soil, the primary minerals will be weathered and genesis of clay minerals can occur. The new formed clay minerals will positively contribute to water retention in soil, since clay minerals with high cation exchange capacity were able to significantly influence soil water retention at a matric potential of -33 kPa (Gaiser et al., 2000). We have also included the potential effect of expected increase in soil organic carbon/matter on plant available water. We placed the caveats in the methodology section (lines 328 to 333) and rephrased the old sentence from line 379, which is now in line 523 to 531.

**RC1: Specific Line by Line Comments 19,21 and 22: Are these ranges 95pct/2 sigma ranges? Lines 147 and 172 suggests that this might be driven by the minimum and maximum harvest rates. What is the central estimate? Abstract should be able to stand on it's own so please specify in parentheticals and provide the median or mean estimate**

Now we explicitly correlate the amount sequestered for each geogenic P supply scenario and AR P demand (high, corresponding to 95th quartile of wood chemistry and low corresponding to 5th quartile of wood chemistry). You can find it in line 20 to 25.

**RC1: 33 and 34: Please put "i.e." and "e.g." phrases in parentheticals**

We did the changes.

**RC1: 59: Suggest authors but "i.e., P, Mg, Ca and K" in the parenthetical before the citation. I also think some more recent work should be cited here and in the next line (60) for controls on atmospheric CO2 over geologic timescales**

We did the changes and cited the study from Singh and Schulze (2015) for weathering as the natural source nutrients (line 63) and the study from Yasunari (2020) for the weathering as control on atmospheric CO2 (line 65)

**RC1: 185: Are the databases sources at most 2 by 2 degrees?**

No, By common convention, fine spatial resolution data are generally upscaled to fit the coarse spatial resolution framework to minimize distortions of location (Pontius, 2000). Besides that downscaling a coarse data is not suggested since information on finer resolution is missing. We added this information in line 202 to line 203.

**RC1: 196: Suggest putting "cf. Wang et al. (2017)..." as a parenthetical**

We accepted the suggestion, you can find it in line 211.

**RC1: 223: Please add citation and average P content in basalt here**

You can find the accepted suggestions in line 258 and 259. Additionally we improved the description on data selection for calculating the descriptive statistics from line 243 to 252.

**RC1: 262: "here estimated" is confusing**

We rephrased the sentence. Now it is explicitly said which equations were used to

obtain the estimated numbers. You can check it in line 341 to 350.

**RC1: 282-283: Is this a citation in the parenthetical?**

We corrected the citation. You can find it in line 443.

**RC1: 307: "under study"?**

We changed the sentence, you can find the changes in line 458 to line 460.

**RC1: 387: please fix grammar of this sentence**

We rephrased it. You can find the new sentence in chapter 4.4 line 537 to line 539.

**RC1: 442: I would suggest citing Uhlig's papers here rather than what I believe from the reference list list is the PhD thesis. The papers are Uhlig et al. (2017, Biogeosciences) and Uhlig and von Blanckenburg (2019, Frontiers in Earth Science)**

We added the suggested citations. You can find them in line 610.

**References**

Ackerer, J., Chabaux, F., Lucas, Y., Clément, A., Fritz, B., Beaulieu, E., Viville, D., Pierret, M. C., Gangloff, S., and Négrel, P.: Monitoring and reactive-transport modeling of the spatial and temporal variations of the Strengbach spring hydrochemistry, Geochimica et Cosmochimica Acta, 225, 17-35, https://doi.org/10.1016/j.gca.2018.01.025, 2018.

Appelo, C. A. J., and Postma, D.: Geochemistry, Groundwater and Pollution, Second Edition, 2005.

Arshad, M., St. Arnaud, R., and Huang, P.: Dissolution of trioctahedral layer silicates by ammonium oxalate, sodium dithionite–citrate–bicarbonate, and potassium pyrophosphate, Canadian Journal of Soil Science, 52, 19-26, 1972.

Beaulieu, E., Lucas, Y., Viville, D., Chabaux, F., Ackerer, P., Goddéris, Y., and Pierret,

M.-C.: Hydrological and vegetation response to climate change in a forested mountainous catchment, Modeling Earth Systems and Environment, 2, 1-15, 10.1007/s40808-016-0244-1, 2016.

Campbell, G. S.: A simple method for determining unsaturated conductivity from moisture retention data, Soil science, 117, 311-314, 1974.

Cánovas, C. R., De La Aleja, C. G., Macías, F., Pérez-López, R., Basallote, M. D., Olías, M., and Nieto, J. M.: Mineral reactivity in sulphide mine wastes: influence of mineralogy and grain size on metal release, European Journal of Mineralogy, 31, 263-273, 10.1127/ejm/2019/0031-2843, 2019.

Gaiser, T., Graef, F., Cordeiro, J., eacute, and Carvalho: Water retention characteristics of soils with contrasting clay mineral composition in semi-arid tropical regions, Soil Research, 38, 523-536, https://doi.org/10.1071/SR99001, 2000.

Gilkes, R., Young, R., and Quirk, J.: Artificial Weathering of Oxidized Biotite: I. Potassium Removal by Sodium Chloride and Sodium Tetraphenylboron Solutions 1, Soil Science Society of America Journal, 37, 25-28, 1973.

Goll, D. S., Moosdorf, N., Hartmann, J., and Brovkin, V.: Climate-driven changes in chemical weathering and associated phosphorus release since 1850: Implications for the land carbon balance, Geophysical Research Letters, 41, 3553-3558, doi:10.1002/2014GL059471, 2014.

Grathwohl, P.: On equilibration of pore water in column leaching tests, Waste Management, 34, 908-918, https://doi.org/10.1016/j.wasman.2014.02.012, 2014.

Hartmann, J., Jansen, N., Dürr, H. H., Kempe, S., and Köhler, P.: Global CO2-consumption by chemical weathering: What is the contribution of highly active weathering regions?, Global and Planetary Change, 69, 185-194, 2009.

Hartmann, J., Moosdorf, N., Lauerwald, R., Hinderer, M., and West, A. J.: Global chemical weathering and associated P-release - The role of lithology, temperature and soil

properties, Chemical Geology, 363, 145-163, 10.1016/j.chemgeo.2013.10.025, 2014.

Hayes, N. R., Buss, H. L., Moore, O. W., Krám, P., and Pancost, R. D.: Controls on granitic weathering fronts in contrasting climates, Chemical Geology, 535, 119450, https://doi.org/10.1016/j.chemgeo.2019.119450, 2020.

Hering, J., and Stum, W.: Oxidative and reductive dissolution of minerals, Rev. Mineral., 23, 427-465, 1990.

Holdren Jr, G. R., and Speyer, P. M.: Reaction rate-surface area relationships during the early stages of weathering—I. Initial observations, Geochimica et Cosmochimica Acta, 49, 675-681, 1985.

Lasaga, A. C., Soler, J. M., Ganor, J., Burch, T. E., and Nagy, K. L.: Chemical weathering rate laws and global geochemical cycles, Geochimica et Cosmochimica Acta, 58, 2361-2386, 1994.

Pierret, M.-C., Cotel, S., Ackerer, P., Beaulieu, E., Benarioumlil, S., Boucher, M., Boutin, R., Chabaux, F., Delay, F., Fourtet, C., Friedmann, P., Fritz, B., Gangloff, S., Girard, J.-F., Legtchenko, A., Viville, D., Weill, S., and Probst, A.: The Strengbach Catchment: A Multidisciplinary Environmental Sentry for 30 Years, Vadose Zone Journal, 17, 10.2136/vzj2018.04.0090, 2018.

Pontius, R.: Quantification error versus location error in comparison of categorical maps, Photogrammetric Engineering and Remote Sensing, 66, 540-540, 2000.

Porder, S., and Hilley, G. E.: Linking chronosequences with the rest of the world: predicting soil phosphorus content in denuding landscapes, Biogeochemistry, 102, 153-166, 10.1007/s10533-010-9428-3, 2011.

Saxton, K. E., and Rawls, W. J.: Soil water characteristic estimates by texture and organic matter for hydrologic solutions, Soil science society of America Journal, 70, 1569-1578, 2006.
Singh, B., and Schulze, D.: Soil minerals and plant nutrition, Nature Education Knowledge, 6, 1, 2015.

Strefler, J., Amann, T., Bauer, N., Kriegler, E., and Hartmann, J.: Potential and costs of Carbon Dioxide Removal by Enhanced Weathering of rocks, Environmental Research Letters, 2018.

Sun, Y., Peng, S., Goll, D. S., Ciais, P., Guenet, B., Guimberteau, M., Hinsinger, P., Janssens, I. A., Peñuelas, J., Piao, S., Poulter, B., Violette, A., Yang, X., Yin, Y., and Zeng, H.: Diagnosing phosphorus limitations in natural terrestrial ecosystems in carbon cycle models, Earth's Future, 10.1002/2016ef000472, 2017.

Sverdrup, H., Warfvinge, P., Blake, L., and Goulding, K.: Modelling recent and historic soil data from the Rothamsted Experimental Station, UK using SAFE, Agriculture, Ecosystems Environment, 53, 161-177, https://doi.org/10.1016/0167-8809(94)00558-V, 1995.

Velbel, M., Taylor, A., and Romero, N.: Effect of temperature on feldspar weathering rates in alpine and non-alpine watersheds, Geol. Soc. Am. Abstr. Program, 1990, 49,

von Liebig, J. F., and Playfair, L. P. B.: Chemistry in its application to agriculture and physiology, JM Campbell, 1843.

Waldbauer, J. R., and Chamberlain, C. P.: Influence of uplift, weathering, and base cation supply on past and future CO 2 levels, in: A history of atmospheric CO2 and its effects on Plants, Animals, and Ecosystems, Springer, 166-184, 2005.

Wang, Y., Ciais, P., Goll, D., Huang, Y., Luo, Y., Wang, Y. P., Bloom, A. A., Broquet, G., Hartmann, J., Peng, S., Penuelas, J., Piao, S., Sardans, J., Stocker, B. D., Wang, R., Zaehle, S., and Zechmeister-Boltenstern, S.: GOLUM-CNP v1.0: a data-driven modeling of carbon, nitrogen and phosphorus cycles in major terrestrial biomes, Geosci. Model Dev., 11, 3903-3928, 10.5194/gmd-11-3903-2018, 2018.

Yasunari, T.: The Uplift of the Himalaya-Tibetan Plateau and Human Evolution: An

Overview on the Connection Among the Tectonics, Eco-Climate System and Human Evolution During the Neogene Through the Quaternary Period, in: Himalayan Weather and Climate and their Impact on the Environment, edited by: Dimri, A. P., Bookhagen, B., Stoffel, M., and Yasunari, T., Springer International Publishing, Cham, 281-305, 2020.

[Figure]

Figure 1: Increases in weathering rates for different temperatures (T), in Kelvin, estimated according to Arrhenius equation ($Arrhenius\ factor\ =\ e^{-Ae/(R*T)}$) for a gas constant R of 0.00831 kJ K$^{-1}$mol$^{-1}$. Average of 6.5%/°C increase for basalt that had an average activation energy (Ae) of 50 kJ/mol, average increase of ~3%/°C for carbonate activation energy of 24 kJ/mol, and average increase of ~10%/°C for a felsic rock type activation energy of 74 kJ/mol.

**Fig. 1.**

---

## Author Response (AR1)

Dear reviewers, we have carefully reviewed the comments and revised the manuscript. We thank you for constructive remarks and took care to include all of them in the manuscript. We added a detailed reply to each point as addressed below.

**Relevant changes:**

- Moved the methodology for estimating the geogenic P supply for AR from supplementary information to main text
  - Merged the figures of geogenic P supply from the supplementary information and added them to the main text
  - Moved the methodology for estimating the impacts on soil hydrology from supplementary information to main text
- 10

15

5

- Created subchapters on supplementary text for N-limited and N-unlimited AR results
- Changed the abstract as suggested by both reviewers
- Inclusion of Results chapter in the main text
- Changed the name of chapter Discussion to Discussion and implications
- Changed the figures according to reviewers suggestions
- Created a new figure for the hypothesis of increased harvest rates for Bioenergy grasses (Supplementary file)

**Reviewers comment**

**Our reply to Daniel Ibarra:**

20

Comments 62: How does this older Gaillardet number compare to those more recently produced by co-author Hartmann's papers and Moon et al. (2014)? I would consider citing all of these sources and giving a range of the CO2 consumption rate by silicate weathering is.

25 In line 67 and 68 we gave the range and considered the suggested authors.

71-76: Work by Porder and Hilley, as well as Waldbauer and Chamberlain (2005) should be cited in this paragraph or previously.

30 The work of Porder and Hilley (2011) is cited in line 80, and Waldbauer and Chamberlain (2005) is cited in line 64.

**97: Have others done this type of analysis or suggested this since von Liebig and Playfair (1843)?**

We rephrased the sentence. According to Liebig's Law of the minimum(von Liebig and Playfair, 1843), biomass productivity
 will be limited by the scarcest nutrient in the soil. This law is widely applied to biological populations and in many ecosystem models. The new sentence is in line 103 to 106.

**108: What about changes in precipitation and runoff?**

40 The weathering rates of minerals are controlled by the following mechanisms: (i) soil pH (Lasaga et al., 1994;Grathwohl, 2014;Arshad et al., 1972); (ii) redox conditions for Fe and Mn minerals (Hering and Stum, 1990;Gilkes et al., 1973;Cánovas et al., 2019); (iii) soil solution composition (Lasaga et al., 1994;Grathwohl, 2014;Hayes et al., 2020) and soil moisture (Sverdrup et al., 1995;Appelo and Postma, 2005); (iv) temperature (Lasaga et al., 1994;Velbel et al., 1990;Hayes et al., 2020);

and (v) grain size/density of exposed structural defects on mineral surfaces (Lasaga et al., 1994;Arshad et al., 1972;Holdren

- 45 Jr and Spever, 1985; Cánovas et al., 2019). In this study we did not account for impact of changes in hydrology on weathering because model projections of hydrology in the future are not reliable. i.e. for most parts of the global model predictions it differs from each other despite similar boundary conditions (e.g. IPCC AR5). Model predictions of temperature change, on the other hand, are more reliable and all models agree on the sign (i.e. warming). Goll et al. (2014) used a simple weathering model that accounts for temperature, runoff and soil thickness controls on weathering, and illustrated that global warming
- 50 was the dominating driver behind changes in weathering using climate change reconstructions from four earth system models. The reconstructions showed large differences with respect to hydrology and changes in global weathering rates to changes in temperature were rather minor. The temperature sensitivity of global weathering rates was significant and rather homogenous among four different climate models. The temperature sensitivity implicitly accounts for changes in hydrology, however it is not able to capture regional differences. Thus we only account for the T changes. This was done earlier in Sun et al. (2017)
- 55 Earth's Future. Besides the large uncertainty with respect to climate change projections, current data availability is insufficient to calibrate a complex global model, which are currently only applied on site to catchment scale (Pierret et al., 2018; Ackerer et al., 2018; Beaulieu et al., 2016). We argue that a simple model of weathering is appropriate. The used model is very simplistic but the used weathering rates were calibrated on 381 catchments in Japan (Hartmann et al., 2009) and describes the chemical
- 60 weathering as a function of runoff and lithology, temperature and soil thickness (Hartmann et al., 2014). The model is increasingly used for global scale studies (Sun et al., 2017;Goll et al., 2014;Wang et al., 2018). We included part of this discussion in line 216 to line 222.

**Relatedly, what are the "Parameters" in yellow in Figure 1? It might help to list them on the figure.**

65

We rearranged Fig.1 and they are explicitly specified there.

169: In addition to Figure 1, a schematic of what is actually being done in the methods own so please specify in parentheticals and provide the median or mean estimate.

70

We rearranged Fig.1, so the used steps are clearer and they refer to the respective chapter sections and used equations.

**191: Again, what about changes in runoff to weathering rate?**

75 See above.

> What is this 9%/degC equivalent to in an Arrhenius style calculation? While temperature increases make sense (kinetics) much of the world weathers chemostatically (equilibirium, see Winnick and Maher (2018, EPSL) and papers by Godsey for example) so changes in soil infiltration rate (P-ET = runoff) and thus soil moisture should also influence these changes.

80

The Arrhenius corresponding average increase is 6.5%/°C for basalt that had an average activation energy of 50 kJ/mol, ~3%/°C for carbonate activation energy of 24 kJ/mol, and ~10%/°C for a felsic rock type activation energy of 74 kJ/mol (Figure 1). Our 9%/°C value is also based on the Arrhenius calculation but represents the average value considering different

85 lithotypes (they had mean apparent activation energies ranging from 50 to 80 kJ/mol and a deviation of  $\pm$  20 kJ/mol, based on field data and laboratory experiments (Goll et al., 2014)). The apparent temperature sensitivity of weathering from Goll et al. (2014) implicitly includes changes in hydrology as it is derived from global model simulations using a model which describes the chemical weathering as a function of runoff and lithology, temperature and soil thickness (Hartmann et al., 2014). As lay down above we omit a more detailed representation

90 of water effects due to high uncertainty with respect to projections of hydrology (IPCC AR5) and as temperature was found to be the dominant driver of changes (Goll et al., 2014).

Figure 1: Increases in weathering rates for different temperatures (T), in Kelvin, estimated according to Arrhenius equation (Arrhenius factor = e-Ae/(R\*T)) for a gas constant R of 0.00831 kJ K-1mol-1. Average of 6.5%/°C increase for basalt that had an average activation energy (Ae) of 50 kJ/mol, average increase of ~3%/°C for carbonate activation energy of 24 kJ/mol, and average increase of ~10%/°C for a felsic rock type activation energy of 74 kJ/mol.

**220-233: What about the grain size of the rock powder be applied? Does that matter at all? I assume it would be based on reactive transport simulations such as those by Maher (2010, 2011) for example. And relatedly how fine the rock is powdered will change the texture described in the next section right?**

Yes, the grain size matters since it will influence on the exposed reactive surface area (generally fine textured basalt powder would have a high reactive surface area than a coarser textured basalt powder) and consequently the weathering rates. However, in our approach we considered that the grain sizes would range in between  $0.6 - 90 \mu m$  (for the fine basalt powder) which is enough to completely dissolve the deployed rock powder after one year (Strefler et al., 2018). For the coarse basalt

- powder ~70% of its granulometry fall into the 0.6 90  $\mu$ m range and from the other 30%, at least 20% would be dissolved in one year (Strefler et al., 2018). The finest grain size we can consider is the clay size which comprehends grains >1  $\mu$ m and <3.9  $\mu$ m. If a basalt powder would have only clay size granulometry, the effects on soil hydraulic conductivity would decrease in average by 37% for deployment amount of 30 kg basalt m2 (for the same deployment amount and for the fine rock powder
- 110 used in our work, the hydraulic conductivity would decrease by 2%). However, the finer the grain gets the higher the energy input for grinding is, which can drastically affect the costs of EW (it can reach up to 500\$/tCO2 sequestered; Streffer et al., 2018). Part of this discussion is included in line 254 to 255. Additionally, we make it clear that we used grain sizes not higher than 90 μm in line 279 to 283 and discussed about the assumptions of full dissolution in chapter 4.4 in line 562.

**115 240-243: Thus, this scenario will be the maximum effect? I think that should be stated explicitly here and elsewhere.**

Thanks, now it is explicitly stated in lines 283, 285, 393, and 394.

100

248 (equation 10) and text surrounding: Is this the 'pedotransfer function' mentioned in the above paragraph? Is the only empirical values the lambda? Further, where does the 1930 in the equation come from? A lot of this appears to be buried in the supplement, maybe it'd be more helpful if it is more explicitly stated that this is from Saxton and Rawls (2006) upfront in this section.

125

This is one of the used pedotransfer functions. The 1930 is a constant (i.e., a regression coefficient) which is dependent on the database used to derive the pedotransfer equation as described in the work from (Campbell, 1974). Now the pedotransfer equations are present in the main text in subchapter 2.6. No, lambda is estimated according to equation 18. The empirical values are the organic matter content and the granulometry of soil and rock powder (equations 15-17).

130

**332: What about other soil types? Is this information in the literature? What is the most common agricultural soil type?**

Considering the extension, Oxisols are the most common agricultural soil type followed by Ultisols. We added the value for Ultisol in line 450 to 453.

**379: But if they are empirical wouldn't those datasets inherently have clay minerals forming in those systems? In general, I think the caveats brought up in this paragraph should be acknowledged earlier in the paper.**

- The used databases probably contained clay minerals in the clay-sized fraction of the soils. But the Saxton and Rawls (2006) equations accounted only for soil texture and soil organic matter content influences on water retention and hydraulic conductivity. The sentence was misleading and we changed it. You can find the improved sentence in line 523 to 531. What was meant by the old sentence was that once the rock powder will be added into the soil, the primary minerals will be wathered and genesis of clay minerals can occur. The new formed clay minerals will positively contribute to water retention in soil, since clay minerals with high cation exchange capacity were able to significantly influence soil water retention at a matric potential of -33 kPa (Gaiser et al., 2000). We have also included the potential effect of expected increase in soil organic
- 145 potential of -33 kPa (Gaiser et al., 2000). We have also included the potential effect of expected increase in soil organic carbon/matter on plant available water. We placed the caveats in the methodology section (lines 328 to 333) and rephrased the old sentence from line 379, which is now in line 523 to 531.

**Specific Line by Line Comments 19,21 and 22: Are these ranges 95%/2 sigma ranges? Lines 147 and 172 suggests that this might be driven by the minimum and maximum harvest rates. What is the central estimate? Abstract should be able to stand on it's own so please specify in parentheticals and provide the median or mean estimate.**

Now we explicitly correlate the amount sequestered for each geogenic P supply scenario and AR P demand (high, corresponding to 95th quartile of wood chemistry and low corresponding to 5th quartile of wood chemistry). You can find it in line 20 to 25.

**33 and 34: Please put "i.e." and "e.g." phrases in parentheticals**

We did the changes.

160

155

**59: Suggest authors but "i.e., P, Mg, Ca and K" in the parenthetical before the citation. I also think some more recent work should be cited here and in the next line (60) for controls on atmospheric CO2 over geologic timescales.**

We did the changes and cited the study from Singh and Schulze (2015) for weathering as the natural source nutrients (line 63) and the study from Yasunari (2020) for the weathering as control on atmospheric  $CO_2$  (line 65)

135

**170 185: Are the databases sources at most 2 by 2 degrees?**

No, By common convention, fine spatial resolution data are generally upscaled to fit the coarse spatial resolution framework to minimize distortions of location (Pontius, 2000). Besides that downscaling a coarse data is not suggested since information on finer resolution is missing. We added this information in line 202 to line 203.

175

**196: Suggest putting "cf. Wang et al. (2017)..." as a parenthetical**

We accepted the suggestion, you can find it in line 211.

**180 223: Please add citation and average P content in basalt here.**

You can find the accepted suggestions in line 258 and 259. Additionally we improved the description on data selection for calculating the descriptive statistics from line 243 to 252.

**185 262: "here estimated" is confusing.**

We rephrased the sentence. Now it is explicitly said which equations were used to obtain the estimated numbers. You can check it in line 341 to 350.

**190 282-283: Is this a citation in the parenthetical?**

We corrected the citation. You can find it in line 443.

**307: "under study"?** 195**

We changed the sentence, you can find the changes in line 458 to line 460.

**387: please fix grammar of this sentence**

200 We rephrased it. You can find the new sentence in chapter 4.4 line 537 to line 539.

442: I would suggest citing Uhlig's papers here rather than what I believe from the reference list list is the PhD thesis. The papers are Uhlig et al. (2017, Biogeosciences) and Uhlig and von Blanckenburg (2019, Frontiers in Earth Science).

205 We added the suggested citations. You can find them in line 610.

210

**Reviewers comment**

**220**

Our reply to anonymous Referee:

Major comments:

- 225 A number of uncertainties are considered in the paper, but it seems that a lot of weight is given to some uncertainties, while other potentially large uncertainties are neglected. For example, only a single CO2/climate scenario is considered and for this particular scenario only results from a single afforestation/bioenergy model is used. What was the reason to choose this particular climate scenario?
- 230 The representative greenhouse concentration pathway 4.5 (RCP4.5) was used (Thomson et al., 2011) since it assumes a emission peak around 2040 and considers that forest lands expand from their present day extent (Thomson et al., 2011). Therefore, this CO2 climate scenario seemed to be the most suited for our purposes since forest expansion under the RCP2.6 is lower than that for the RCP4.5 and RCP8.5 scenarios (van Vuuren et al., 2011). The RCP8.5 scenario assumes no climate policy being adopted and consequently the expansion on forest cover does not occur (van Vuuren et al., 2011). There are at
- least 9 different terrestrial ecosystem models for coupled C-nutrient cycle. We selected the JSBACH since it considers the same level of competition between plants and decomposing microbes for N supply, while other numerical models prioritize immobilization or plant growth (Achat et al., 2016). Based on simulations from Parida (2011), prioritizing the Plant N uptake is unrealistic because soil microbes are more competitive for soil N as compared by plants. Besides that, the selected AR model considers the impacts on biomass productivity due to natural N supply or to N fertilization at a global scale AR deployment (we considered this comment in lines 133 to 148 for AR).
- Since we want to point out the influence of plant nutrition on biomass growth, using other numerical models would not change our final message (that is: Geogenic nutrient supply can limit biomass growth and consequently reduce the sequestered CO2 potential of large scale AR).
- This is seen for AR since simulations using the RCP4.5 land pattern that accounted for the RCP8.5 scenario for tree harvest
   rates and atmospheric CO2 concentrations. The resulting output showed similar areas of forest growth and higher biomass productivity (Sonntag et al., 2016). However, these results did not consider natural N supply and consequently biomass productivity would be N limited, which would decrease the amount of Carbon within Biomass. Similar areas can also be observed in the study from Yousefpour et al. (2019) in Figure 2c for different RCP scenarios.
- The selected bioenergy model can minimize total costs of production. It also accounts for vegetation composition and distribution for both natural and agricultural ecosystems. Additionally, it considers socio-economic conditions of a system. We selected the RCP4.5 scenario for bioenergy grasses (BG) to keep the MAgPIE simulations corresponding to the JSBACH simulations (we considered this comment in lines 155 to 156 for BG). In the case of BG, to keep high yields or maintain a selected harvest rate, it will be necessary to replenish the exported nutrients by harvest. Therefore, external source of nutrients will be necessary. Accounting for another model rather than MAgPIE would probably change the estimated minimum and
- 255 maximum harvest rates for Bioenergy grasses, which would impact the necessary amount of rock powder to replenish the exported nutrients due to harvest rates. Thus, we added a hypothetical scenario assuming that the estimated maximum harvest rate by MAgPIE could be increased by one order of magnitude, even though this is unlikely to occur. Adding a scenario accounting for decrease in harvest rates is not necessary, since it is already presented in Fig. 8 from the main text by the filled horizontal lines. However, decreasing the minimum a harvest rate by a factor of two or three also represent a decrease in the minimum amount of rock powder necessary to replenish the exported nutrients by harvest.
- Since the core messages of our study are: (i) Biomass productivity is limited by geogenic P supply; (ii) EW is an alternative for supplying nutrients besides the potential to keep a net positive  $CO_2$  balance; and (iii) the effect of EW on soil hydrology can be neglected in some parts of the world, but EW has the potential to alleviate water stress, at some extent, in areas that drought occurs. Accounting for other models either for AR or bioenergy grasses would not change the general message.

**A discussion on the applicability of the conclusions drawn from the paper to alternative scenarios should be included.**

We included the discussion on chapter 4.1 in lines 461 to 467 considering the results from others AR model that assumes
 RCP8.5 for an RCP4.5 land evolution scenario. And we have also acknowledged your point (that "even using only one model induces uncertainty" the general message of the study would not change) in the last sentence.

We included a discussion for the harvest rates obtained from the MAgPIE simulations. In this discussion we assume a hypothetical one order of magnitude increase in the maximum harvest rate. The discussion is presented on chapter 4.2 in line 488 to 498.

**For the carbon sequestration from afforestation you mention (line 260) that the available estimates of carbon uptake vary between 0.3 and 3.3 GtC/a, while in the paper a value of 2 GtC/a is used.**

- **280** This is the value for the AR scenario from Kracher (2017), the same model shows a carbon sequestration of ~2.4 Gt C  $\alpha$ -1 If N supply is unlimited. We have shown that it can fall to ~1.3 Gt C  $\alpha$ -1 if geogenic P supply scenario one for mean P content within wood and leaves is selected. This number would change for another AR scenario. But the main message is that the estimated C sequestration by biomass on terrestrial carbon cycle models can fall if nutrient supply is accounted for.
- 285 On the other hand, a lot of weight in the uncertainty is given to e.g. the P concentration in basalt (5-95th percentiles). To my understanding it seems that this uncertainty is not so relevant for the present study, as I assume that for the use in EW basalt with relatively high P concentrations could in principle be selected. Would the interquartile range possible provide a more appropriate measure of the uncertainty in this parameter?
- 290 For EW, we need to firstly know rock mineralogical composition and petrography. Therefore, it is more interesting a basalt with high pyroxene group minerals content (especially the ones rich in Ca and Mg like Diopside) since these minerals would weather more rapidly (cf. Table 1 at Hartmann et al. (2013)) and less olivine or sulfide minerals content (Olivine can have high content of Nickel and Chromium that are trace elements problematic for agriculture (Edwards et al., 2017); Sulfide minerals can cause acid rock drainage if pyrite (a sulfide mineral) concentration is within 1% or 2% (Earle, 2018)). As an
- 295 example, Alkali basalt can have P concentration >3000 ppm (Porder and Ramachandran, 2013), but it is rich in olivine (John, 2001; Irvine and Baragar, 1971). Therefore, in this study we have adopted a more strict data selection from the EarthChem database. We selected only the rocks exclusively named as rhyolite, dacite, andesite, and basalt. This resulted in 2985 samples for rhyolite, 3008 samples for dacite, 11099 samples for andesite, and 23816 samples for basalt. Comparing our P concentration values to other works, we see that the median P value of 916 ppm from Porder and Ramachandran (2013) is
- 300 higher than ours (500 ppm of P). This occurs because Porder and Ramachandran (2013) did a broader classification, resulting in 97895 used samples, and neglected possible unwanted side-effects of trace minerals in basalt mineralogy. Therefore, the selected quartiles for rock chemistry (either for Basalt or other rock used in our study) are a conservative estimation and assuming interquartile values would decrease even more the consideration of potential rock sources for EW.

**305 In general in the paper it is difficult to immediately associate the uncertainty ranges to uncertainty sources.**

We re-structured the paper. The results are discussed in a separate section. We also renamed the old section "results and discussion" to **Discussion and implications**, which contain the discussion for the presented results and implications to rock powder deployment.

310

- 315 For AR it seems that the largest uncertainty is related to the geogenic P-supply, with scenario two showing basically no P limitation. Is there really no observation-based evidence to suggest that one or the other scenario is more realistic? This is a fundamental question for the purpose of the paper, because if P limitation is not an issue the benefit of EW in these areas will be limited to the direct CO2 consumption by weathering.
- Basically, the uncertainties for the AR scenario are from biomass P demand and the geogenic P supply, with the later influencing the most our results as it was seen. Unfortunately, given current understanding of bioavailability of soil P and soil P estimates uncertainties are large with respect to how much P is available to support future plant growth (detailed analysis and discussion is found in Sun et al. (2017)) and thus AR and BG. Mineral P is likely limiting biomass production in European forests today (Jonard et al., 2015), tropical forest (Turner et al., 2018), boreal forests (Shinjini et al., 2018), as well as agricultural areas (e.g., Ringeval et al., 2019 in discussion;Kvakić et al., 2018). This situation is likely to deteriorate in the future. Therefore, considering that the inorganic and organic labile P pools will be completely available for tree nutrition is
- unlikely to occur. Thus, Geogenic supply Scenario 2 is a very optimistic assumption that might not correspond to reality based on the already observed P limitation on different ecosystems (Elser et al., 2007). However, we cannot rule out that gradual shifts in soil organic P fractions occur, which make comparable amounts of P as in scenario 2 available over time, We therefore opted to show both scenarios as these are a major source of uncertainty with respect to P effects on future plant growth (as has been demonstrated by Sun et al. (2017)).

In the main text there are many references to supplementary materials. References to supplementary material should be limited as much as possible for a better readability of the paper, considering also that papers in Biogeosciences are not subject to strict length limitations. Since the main paper contains only relatively few figures, I would suggest to move some of the figures from the supplementary to the main paper. For example Figs S1,S2,S3 could be merged into one figure and added to the paper. It would help to get an idea of the numbers that could then be more easily compared with e.g. the P gap in Fig. 2. I would also strongly suggest to add Fig. S8 to the main paper.

**340** We appreciate your suggestions and have considered them. Now the Figs. S1 to S3 is the Fig. 3 in the mains text. The Fig. S8 now is the Fig. 4 in the main text (which also include the estimated P gaps for the AR scenario).

The results for the N-unlimited scenario, which are not discussed in the main text, could be presented in a separate section in the supplementary material. It is very confusing to find figures from N-unlimited experiments mixed inbetween N-limited figures.

Now in the Chapter B from the supplementary material we have the results for the AR N-limited for the  $5^{th}$  and  $95^{th}$  quartile of wood P content in the subchapter 'i'. The AR N-unlimited results are presented within the subchapter 'ii'.

350 Minor comments:

Figures should be numbered progressively according to where they are referenced in the text (i.e. Fig. 1 should be referenced before Fig. 2). This is valid also for figures in the supplementary.

We did the suggestion. You can find it in the supplementary file S1.docx line 47 to 53.

**Line 21-22: you need to mention what scenario you are considering here, otherwise the numbers make no sense since they will be strongly dependent on future CO2/climate evolution.**

360

Now we explicitly correlate the amount sequestered for each geogenic P supply scenario and AR P demand (high, corresponding to 95th quartile of wood chemistry and low corresponding to 5th quartile of wood chemistry). Line 20 to line 25.

**365 Line 23: 'K' not defined here**

Now it is defined (line 26).

**Line 22: it would be helpful if the same unit would be used, either GtC or GtCO2. Otherwise it is difficult to compare the numbers.**

Now they are in the same unit (GtC) line 25.

**Line 95: '...we discuss only the N-limited AR scenario...' 375**

We did the change, you can see it in line 102.

**Line 119: please check this sentence**

380 The sentence was deleted.

**Line 147: are these numbers global averages for the areas with bioenergy plantations?**

No, they are global minimum and maximum. They are necessary since for bio-energy crops the amount of exported nutrients
by harvest need to be replenished to keep with the specific harvest rate. The sentence was reformulated, you can see it in line 164.

**Line 186: nearest neighbour interpolation does not seem to be a very good interpolation choice for very high resolution input data (e.g. geogenic P release rates). A single 1 km2 value will not be very representative for a 2x2 cell.**

390

No, according to Christman and Rogan (2012) the nearest-neighbor scaling method can keep with the overall proportions of the original fine resolution map. This is because the interpolant exhibits the smallest variation of the interpolant function while meeting the measured data. Since we are interested in P concentrations (supplied by weathering, atmospheric deposition, or different soil pools) the degree of discontinuity is very high (i.e., they are spatially and temporally heterogeneous) and using a more robust interpolation algorithm (i.e., Cubic-Spline) could result in new minima and maxima values in the original maps, which would be a wrong result. If the Best Linear Unbiased Estimator (i.e., Kriging) algorithm is selected, it would be necessary to know the proper semivariogram function or assume one based on tests to check its appropriateness or on the uncertainty estimation. Therefore, to preserve the limits of different geogenic P release rates represented by the fine resolution map from Hartmann et al. (2014) and due to its simplicity compared to the other interpolation algorithms, we selected the

400 nearest-neighbor interpolation method. You can see these justifications in line 202 to line 204.

**Line 425: where does this number come from?**

We rephrased the sentence. Now you can find it in line 592.

(belegited beter all (19) And ( The potential changes in soil hydraulic properties, due to the application of a fine basalt texture (15.6% clay, 83.8% silt, and 820 0.6% fine sand) or a coarse basalt texture (15.6% clay, 53.8% silt, and 30.6% fine sand) were estimated as a function of rock powder deployment for soils corresponding to P gap areas from the N-unlimited AR scenario. According to the international organization for standardization, the man-made materials can be classified according to their grain sizes; therefore, here the clay comprehends grain diameters  $\leq 2 \,\mu$ m, silt comprehends grain diameter  $2 - 63 \,\mu$ m, and fine sand comprehends 63 - 200 µm (ISO 14688-1:2002), but since full dissolution is assumed, the ground basalt fine sand encompass grain sizes of 825 diameter 63 - 90 µm remaining withing the ISO 14688-1:2002 classification. The N-unlimited AR scenario was selected since it would have the highest P deficiencies requiring more rock powder to cover the P gaps (i.e., it represents the maximum effect). The estimations are for a homogeneous mixture of rock powder and topsoil depth of 0.3 m. Downward transport of fine-grained material is neglected for simplification. The considered values represent upper limits of rock powder application. The impacts on plant available water (PAW) is given by the difference between water content at a pressure head of -33 kPa 830 (Eq. (11)) and -1500 kPa (Eq. (10)), while the impact on soil hydraulic conductivity is given by (Eq. (14); Saxton and Rawls, 2006):

| $\theta_{1500} = \theta_{1500t} + (0.14 \times \theta_{1500t} - 0.02),$                               | ( 10)    |
|-------------------------------------------------------------------------------------------------------|-----------------|
| $\theta_{33} = \theta_{33t} + (1.283 \times (\theta_{33t})^2 - 0.374 \times (\theta_{33t}) - 0.015),$ | ( 11)    |
| $\theta_{(S-33)} = \theta_{(S-33)t} + (0.636 \times \theta_{(S-33)t} - 0.107).$                       | ( 12)    |
| $\theta_{S} = \theta_{33} + \theta_{(S-33)} - 0.097 \times S + 0.043_{2}$                             | ( 13)    |
| $K_{S} = 1930 \times (\theta_{S} - \theta_{33})^{(3-\lambda)},$                                       | <del>(10)</del> |

wherewith:

| $\theta_{1500t} = -0.024 \times S + 0.487 \times C + 0.006 \times OM + 0.005 \times (S \times OM) - 0.013(C \times OM) + 0.068(S \times C) + 0.006(S \times$                                                                                                                                                                                                                                                                                                                                      | (15) |
|---------------------------------------------------------------------------------------------------------------------------------------------------------------------------------------------------------------------------------------------------------------------------------------------------------------------------------------------------------------------------------------------------------------------------------------------------------------------------------------------------------------------------------------------------------------------------------------------------------------------------------------------------------------------------------------------------------------------------------------------------------------------------------------------------------------------------------------------------------------------------------------------------------------------------------------------------------------------------------------------------------------------------------------------------------------------------------------------------------------------------------------------------------------------------------------------------------------------------------------------------------------------------------------------------------------------------------------------------------------------------------------------------------------------------------------------------------------------------------------------------------------------------------------------------------------------------------------------------------------------------------------------------------------------------------------------------------------------------------------------------------------------------------------------------------------------------------------------------------------------------------------------------------------------------------------------------------------------------------------------------------------------------------------------------------------------------------------------------------------------------------------------------------------------------------------------------------------------------------------------------------------------------------------------------------------------------------------------------------------------------------------------------------------------------------------------------------------------------------------------------------------------------------------------------------------------------------------------------------------------------------------------------------------------------------------------------------------------------------------------------------------------------------------------------------------------------------------------------------------------------------------------------------------------------------------------------------------------------------------------------------------------------------------------------------------------------------------------------------------------------------------------------------------------------------------|-------------|
| 0.031_                                                                                                                                                                                                                                                                                                                                                                                                                                                                                                                                                                                                                                                                                                                                                                                                                                                                                                                                                                                                                                                                                                                                                                                                                                                                                                                                                                                                                                                                                                                                                                                                                                                                                                                                                                                                                                                                                                                                                                                                                                                                                                                                                                                                                                                                                                                                                                                                                                                                                                                                                                                                                                                                                                                                                                                                                                                                                                                                                                                                                                                                                                                                                                                |             |
| $\theta_{33t} = -0.251 \times S + 0.195 \times C + 0.011 \times OM + 0.006 \times (S \times OM) - 0.027 \times (C \times OM) + 0.452(S \times C) + 0.0000 \times (S \times OM) + 0.00000 \times (S \times OM) + 0.$ | (16)        |
| 0.299.                                                                                                                                                                                                                                                                                                                                                                                                                                                                                                                                                                                                                                                                                                                                                                                                                                                                                                                                                                                                                                                                                                                                                                                                                                                                                                                                                                                                                                                                                                                                                                                                                                                                                                                                                                                                                                                                                                                                                                                                                                                                                                                                                                                                                                                                                                                                                                                                                                                                                                                                                                                                                                                                                                                                                                                                                                                                                                                                                                                                                                                                                                                                                                                | ± .7        |
| $\theta_{(S-33)t} = 0.278 \times S + 0.034 \times C + 0.022 \times OM - 0.018 \times (S \times OM) - 0.027 \times (C \times OM) - 0.584 \times OM - 0.018 \times (S \times OM) - 0.027 \times (C \times OM) - 0.000 \times OM - 0.0000 \times OM - 0.000 \times OM - 0.000 \times OM - 0.000 \times OM - 0.000 \times OM - 0.0000 \times OM - 0.0000 \times OM - 0.000 \times OM - 0.0000 \times$                                                                                                                                                                                                                             | (17)        |
| $(S \times C) + 0.078_{L}$                                                                                                                                                                                                                                                                                                                                                                                                                                                                                                                                                                                                                                                                                                                                                                                                                                                                                                                                                                                                                                                                                                                                                                                                                                                                                                                                                                                                                                                                                                                                                                                                                                                                                                                                                                                                                                                                                                                                                                                                                                                                                                                                                                                                                                                                                                                                                                                                                                                                                                                                                                                                                                                                                                                                                                                                                                                                                                                                                                                                                                                                                                                                                            |             |

[revised manuscript text omitted]